# Taming Diffusion Prior for Image Super-Resolution with Domain Shift SDEs

**Qinpeng Cui**[*12], **Yixuan Liu**[1], **Xinyi Zhang**[2], **Qiqi Bao**[2], **Qingmin Liao**[2]
**Li Wang**[1], **Tian Lu**[1], **Zicheng liu**[1], **Zhongdao Wang**[†2], **Emad Barsoum**[1]
[1]Advanced Micro Devices Inc.     [2]Tsinghua University
{qinpeng.cui, yixuan.liu, li.wang, lu.tian, zicheng.liu, ebarsoum}@amd.com;
{cqp22, xinyi-zh22, bqq19, liaoqm, wcd17}@tsinghua.edu.cn

## Abstract

Diffusion-based image super-resolution (SR) models have attracted substantial interest due to their powerful image restoration capabilities. However, prevailing diffusion models often struggle to strike an optimal balance between efficiency and performance. Typically, they either neglect to exploit the potential of existing extensive pretrained models, limiting their generative capacity, or they necessitate a dozens of forward passes starting from random noises, compromising inference efficiency. In this paper, we present DoSSR, a **Do**main **S**hift diffusion-based SR model that capitalizes on the generative powers of pretrained diffusion models while significantly enhancing efficiency by initiating the diffusion process with low-resolution (LR) images. At the core of our approach is a domain shift equation that integrates seamlessly with existing diffusion models. This integration not only improves the use of diffusion prior but also boosts inference efficiency. Moreover, we advance our method by transitioning the discrete shift process to a continuous formulation, termed as DoS-SDEs. This advancement leads to the fast and customized solvers that further enhance sampling efficiency. Empirical results demonstrate that our proposed method achieves state-of-the-art performance on synthetic and real-world datasets, while notably requiring *only 5 sampling steps*. Compared to previous diffusion prior based methods, our approach achieves a remarkable speedup of 5-7 times, demonstrating its superior efficiency. Code: https://github.com/AMD-AIG-AIMA/DoSSR

## 1   Introduction

Image super-resolution (SR) is a classical task in computer vision that involves enhancing a low-resolution (LR) image to create a perceptually convincing high-resolution (HR) image [28]. Traditionally, this field has operated under the assumption of simple image degradations, such as bicubic down-sampling, which has led to the development of numerous effective SR models [6, 25, 57, 12]. However, these models often fall short when confronted with real-world degradations, which are typically more complex than those assumed in academic settings. Recently, diffusion models has emerged as a pivotal research direction in real-world SR, using their robust generative capabilities to enhance perceptual quality. This shift highlights their superior performance in practical applications.

Currently, diffusion-based SR strategies can be broadly categorized into two approaches. The first approach leverages large-scale pretrained diffusion models (*e.g.*, Stable Diffusion [41]) as generative prior, using LR images (or preprocessed LR images) as *conditional inputs* to generate HR images [45, 27, 51]. Despite achieving remarkable results, it exhibits low inference efficiency,

---

[*]Work done during an internship at AMD.
[†]Corresponding author.

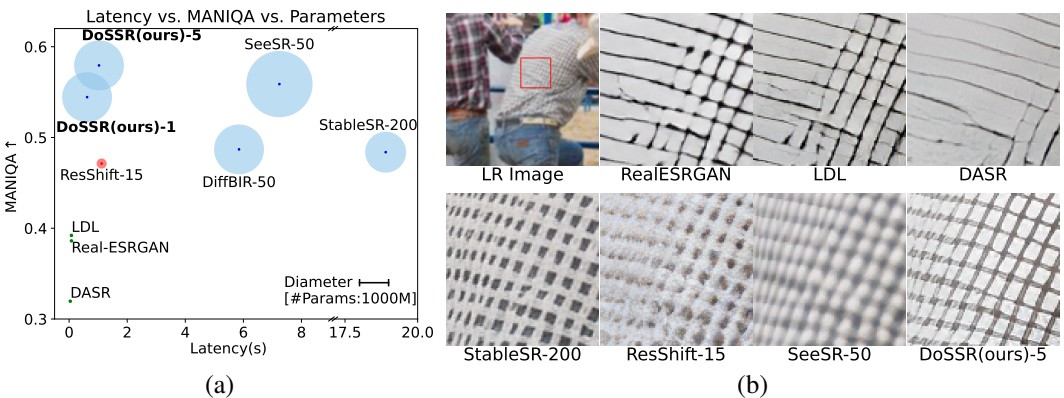

Figure 1: (a) Latency, MANIQA, and complexity of model comparison on *RealLR200* [51] dataset in x4 SR task (for 128×128 LR images). (b) Qualitative comparisons of DoSSR and recent state-of-the-art methods on one typical real-world example. For diffusion-based methods, the suffix "-N" appended to the method name indicates the number of inference steps. Zoom in for a better view.

as the inference starting point is a random Gaussian noise instead of the LR image. Although techniques such as sampler optimization [33, 38, 30, 9, 22] or model distillation [32, 37] have been proposed to mitigate this issue, they inevitably compromise SR performance. The second approach involves redefining the diffusion process and retraining a model from scratch for the SR task [53, 36]. Consequently, the generative prior from pretrained diffusion models is not leveraged. ResShift [53], as a typical representative, revises the forward process of DDPM [16] to better accommodate the SR task. By starting from LR rather than Gaussian noise, it improves inference efficiency. However, its modification of the diffusion pattern, which deviates significantly from existing noise schedules in diffusion models, hinders its integration with large-scale pretrained diffusion models for leveraging their generative prior. The diffusion generative prior has been proven to be highly beneficial for SR tasks [45], enabling models to transcend the limitations of knowledge learned solely from the training dataset, thereby equipping them to handle various complex real-world scenarios. Thus, crafting a diffusion process tailored for the SR task that also remains compatible with established diffusion prior presents a significant challenge.

To tackle this challenge, we propose DoSSR, a **Do**main **S**hift diffusion-based SR model. We initially view the SR task as a gradual shift from the LR domain to the HR domain, describing this transition with a linear equation, which is called *domain shift equation*. Then, we combine this domain shift equation with existing diffusion equations, facilitating the fine-tuning of large-scale pretrained diffusion models to harness diffusion prior effectively. Moreover, by carefully designing a shifting sequence, inference can begin from LR images rather than Gaussian noises, thereby boosting inference efficiency. To further enhance efficiency, we employ sampler optimization techniques, extensively explored in image generation [38, 30, 9], but not previously tailored for diffusion-based SR tasks. Specifically, we expand the customized diffusion equation from discrete to continuous, enabling its formulation as stochastic differential equations (SDEs). We subsequently present the corresponding backward-time SDE as Domain Shift SDE in the reverse process and provide an exact formulation of its solution. Based on our formulation, we customize fast solvers for sampling. Experimental results demonstrate that our method achieves superior or comparable performance compared to current state-of-the-art methods on both synthetic and real-world datasets, ***with only 5 sampling steps***, striking an optimal balance between efficiency and effectiveness. Furthermore, our approach can match the performance of previous methods ***even with just a single step***.

In summary, the main contributions of our work are as follows:

- We propose a novel diffusion equation, which models SR from the perspective of domain shift, enabling inference to start from LR images and leveraging diffusion prior to ensure both efficiency and performance.
- We further propose the SDEs to describe the process of domain shift and provide an exact solution for the corresponding reverse-time SDEs. Based on the solution, we design customized fast samplers, resulting in even higher efficiency, thereby achieving the state-of-the-art efficiency-performance trade-off.

## 2 Related work

**Neural Network-based Super-Resolution.** Neural network-based methods have emerged as the dominant approach in image SR tasks. The introduction of convolutional neural networks (CNNs) and Transformer architecture, with the primary focus on network architecture design [12, 10, 25, 26, 58, 21, 56, 57], have demonstrated superior performance over traditional methods. This improvement is facilitated by the introduction of residual blocks, dense blocks and attention mechanisms. These methods primarily aim for better image fidelity measures such as PSNR and SSIM [49] indices, therefore, they often yield over-smoothed outcomes. To enhance visual perception, Generative adversarial network (GAN)-based SR methods have been developed. By incorporating adversarial loss during training, many SR models [13, 23, 17] can generate perceptually realistic details, thereby enhancing visual quality. To further study SR problems in real-world scenarios, some studies [54, 46, 24] have proposed simulating the intricate real-world image degradation process through random combinations of fundamental degradation operations. Despite the remarkable advancements, GAN-based SR methods can introduce undesirable visual artifacts.

**Diffusion-based Super-Resolution.** Recently, diffusion-based SR methods [35, 36, 8, 7, 45, 27] have demonstrated excellent performance, especially in terms of perceptual quality. These methods can generate more authentic details while avoiding unpleasant visual artifacts like GAN-based methods. Current diffusion models for super-resolution can be broadly categorized into two main approaches. The first approach involves leveraging large-scale pretrained diffusion models, such as Stable Diffusion [41], as prior, and then using LR images as conditional inputs to generate HR images. StableSR [45] and DiffBIR [27] represent representative works that leverage diffusion prior, leading to enhanced fidelity when conditioning on LR or preprocessed LR. SeeSR [51] and CoSeR [42] demonstrate that extracting semantic text information from LR images as additional control conditions for the T2I model helps improve performance. The second approach involves redefining the diffusion process and retraining a model from scratch for SR [18, 36]. To address the slow inference speed issue of diffusion-based SR methods, ResShift [53] constructs a Markov chain that transitions between HR and LR images by shifting residuals between them, enabling accelerated sampling. SinSR [48] proposed a method of distilling ResShift to achieve comparable performance in a single step. Despite the remarkable advancements achieved by ResShift and SinSR, they necessitate retraining from scratch for SR tasks (or further distillation) and are unable to leverage diffusion prior. Therefore, improving the inference efficiency while leveraging the potential of large-scale pretrained diffusion models to assist SR requires thorough investigation, which is the goal of this work.

## 3 Methodology

We aim to optimize the balance between efficiency and performance in diffusion-based super-resolution (SR) models. Our approach is grounded in two key principles: First, initiating inference from LR images rather than noise; Second, effectively harnessing pretrained diffusion prior. In Section 3.1, we introduce a novel diffusion equation designed to fulfill both criteria simultaneously. Subsequently, in Section 3.2, we extend this diffusion process to continuous scenarios, formulating it through Stochastic Differential Equations (SDEs). Building on these SDEs, we develop an efficient solver detailed in Section 3.3, further enhancing inference efficiency.

### 3.1 Diffusion Process with Domain Shift

Our goal is to characterize the shift from the source domain to the target domain as a diffusion process. In the task of SR, the distribution of LR images $p_{\texttt{data}}(\hat{x}_0)$ represents the source domain, while the distribution of HR images $p_{\texttt{data}}(x_0)$ represents the target domain. Firstly, we conceptualize domain shift as a gradual transition from the source domain to the target domain through a linear drift coefficient $\eta_t$, the domain shift equation is formulated as

$$\mathcal{D}(\hat{x}_0, x_0) = \eta_t \hat{x}_0 + (1 - \eta_t)x_0,\ 0 \le \eta_t \le 1,\ t = 1, 2, \cdots, T, \tag{1}$$

where shifting sequence $\{\eta_t\}_{t=1}^T$ monotonically non-decreases with timestep $t$. In order to enable linear combination, we can interpolate $\hat{x}_0$ to match the same dimensions as $x_0$ if necessary. Secondly, we combine this domain shift with the diffusion equation. To integrate with pretrained diffusion models, we adopt the most commonly used diffusion scheme from DDPM [16] and express the

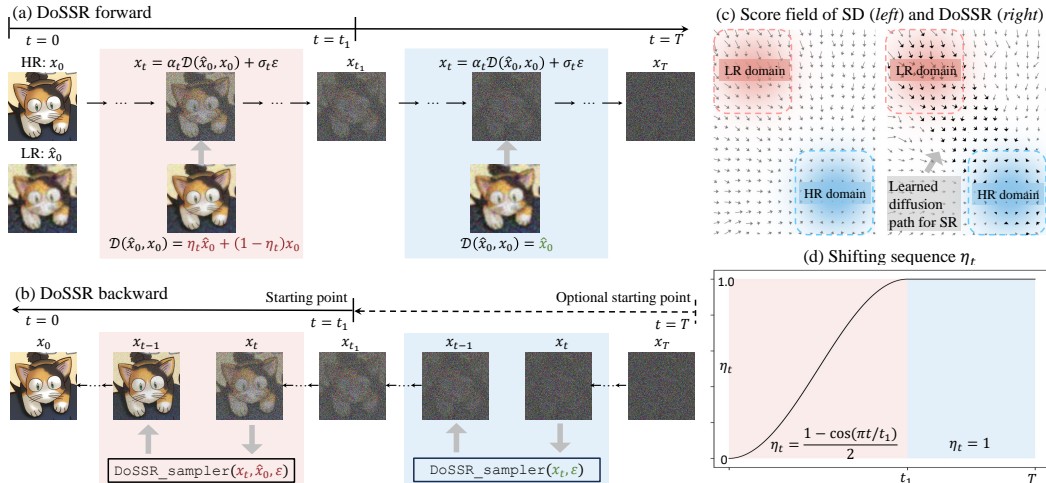

Figure 2: Illustration of the proposed diffusion process with domain shift. (a) In the forward process, we merge the gradual shift from HR to LR domain with standard diffusion process. (b) In the reverse process, we initiate inference from LR domain ($t = t_1$) and use our fast sampler to generate SR images. (c) Comparison of the estimated score between SD and DoSSR. DoSSR inherits the capability of SD in ambient space and enhances learning a pathway from LR to HR domain. (d) The design of the shifting sequence which enables us to initiate inference from $t_1$.

formula of marginal distribution at any timestep $t$ as follows:

$$q(\boldsymbol{x}_t|\boldsymbol{x}_0, \hat{\boldsymbol{x}}_0) = \mathcal{N}(\boldsymbol{x}_t; \alpha_t \mathcal{D}(\hat{\boldsymbol{x}}_0, \boldsymbol{x}_0), \sigma_t^2 \boldsymbol{I}), \ t = 1, 2, \cdots, T, \tag{2}$$

where $\alpha_t, \sigma_t \geq 0$ and $\alpha_t^2 + \sigma_t^2 = 1$, $\boldsymbol{I}$ is the identity matrix. Based on our proposed marginal distribution Eq. (2), we demonstrate the transition distribution as follows:

$$q(\boldsymbol{x}_t|\boldsymbol{x}_{t-1}, \hat{\boldsymbol{x}}_0) = \mathcal{N}(\boldsymbol{x}_t; \frac{\alpha_t}{\alpha_{t-1}}\boldsymbol{x}_{t-1} + \alpha_t(\eta_t - \eta_{t-1})\boldsymbol{e}_0, 1 - \frac{\alpha_t^2}{\alpha_{t-1}^2}\boldsymbol{I}), \ t = 1, 2, \cdots, T, \tag{3}$$

where $\boldsymbol{e}_0 = \hat{\boldsymbol{x}}_0 - \boldsymbol{x}_0$ is the residual between the source and target domain.

**Relation to DDPM [16].** The formulation of Eq. (2) is based on the DDPM [16] forward process, with a crucial difference lying in its mean $\alpha_t \mathcal{D}(\hat{\boldsymbol{x}}_0, \boldsymbol{x}_0)$ instead of $\alpha_t \boldsymbol{x}_0$. This integration encapsulates the domain shift within the variation of its mean, while the diffusion process with added noise maintains consistency with it, thereby smoothing this transformation. Meanwhile, it enhances the diffusion model to learn the pathway from the source domain to the target domain. For an intuitive understanding, we plot and compare the score function $\nabla_{\boldsymbol{x}} \log q_t(\boldsymbol{x}_t)$ learned by a vanilla Stable Diffusion (SD) model and DoSSR in Fig. 2(c). While SD learns reasonable score field in the whole space, DoSSR inherits its capability in the ambient space and further learns more accurate scores along the path between LR and HR domains. Therefore sampling efficiency is improved. Furthermore, the choice of $\alpha_t$ and $\sigma_t$, referred to as the *noise schedule*, follows the existing pretrained diffusion model, allowing us to fine-tune it rather than training it from scratch.

**Relation to ResShift [53].** The form of equation Eq. (3) suggests that this shift essentially constructs a Markov chain in a manner similar to that described in ResShift [53]. However, the equation constructed by ResShift adopts an entirely different noise schedule compared to the pretrained diffusion model. This makes it difficult to apply pretrained diffusion models for subsequent fine-tuning, necessitating training from scratch instead. Therefore, it is unable to utilize the diffusion prior, thereby limiting the model's performance. See Appendix A.7 for detailed theoretical differences from ResShift. In Appendix C.1, we present experimental results on ImageNet [11] showing that DoSSR uses two orders of magnitude less training data than ResShift while achieving superior performance.

**Shifting Sequence.** The parameter $\eta_t$ plays a crucial role in guiding the diffusion process, serving as a bridge between the source and target domains. Specifically, $\eta_t = 1$ represents standard diffusion

forward perturbations in the source domain, whereas $\eta_t = 0$ corresponds to the target domain. The transition between these domains occurs for $0 < \eta_t < 1$, indicating a domain shift. To effectively utilize the diffusion prior, we adopt the noise schedule from DDPM. This adoption dictates that as $t$ approaches the final time step $T$, the scale parameter $\alpha_T$ tends towards zero, and the distribution $q(\boldsymbol{x}_T)$ approximates a standard Gaussian, $\mathcal{N}(\mathbf{0}, \boldsymbol{I})$. To retain prior information from the source domain while shortening the diffusion path, we set $\eta_t = 1$ for $t \in [t_1, T]$, as defined by:

$$\eta_t = \frac{1 - \cos(\pi \frac{t}{t_1})}{2} \text{ if } t \in [0, t_1], \quad \eta_t = 1 \text{ if } t \in [t_1, T]. \tag{4}$$

The advantage of such a setting lies in the fact that during the reverse process, the values of $\boldsymbol{x}_t$ for $t \in [t_1, T]$ are known and can be obtained through the forward process Eq. (2). Consequently, the inference does not need to start from time step $T$, but can commence at $t_1$, thereby preserving the prior information of the source domain while enhancing the efficiency of inference. An overview of the impact of $\eta_t$ is presented in Fig. 2.

## 3.2 Diffusion DoS-SDEs

To improve the efficiency of inference in diffusion models, many prior works [30, 31, 9] have designed efficient samplers by solving the diffusion SDEs. Therefore, in this section, we extend the aforementioned discrete shift process to an SDE for description, in preparation for designing efficient samplers in the following section. Specifically, inspired by the work of [40], we generalize this finite shift process further to an infinite number of noise scales, such that the data distribution of domain shift evolves according to an SDE as noise intensifies. Then we provide the corresponding reverse-time SDE and elucidate the training of diffusion models from the perspective of score matching [39]. Next, we will elaborate extensively on how to describe diffusion models using SDEs.

**Forward Process.** Expanding the time variable $t$ in Eq. (2) to a continuous range, $t \in [0, T]$, we have that $\alpha_t, \sigma_t, \eta_t$ are differentiable functions of $t$ with bounded derivatives. Furthermore, Song *et al.* [40] have demonstrated that the diffusion process can be modeled as the solution to an Itô SDE and we formulate the SDE as follows:

$$d\boldsymbol{x}_t = [f(t)\boldsymbol{x}_t + h(t)\hat{\boldsymbol{x}}_0]dt + g(t)d\boldsymbol{w}_t, \ \boldsymbol{x}_0 \sim q_0(\boldsymbol{x}_0), \tag{5}$$

where $\boldsymbol{w}_t$ is the standard Wiener process, and $q_0(\boldsymbol{x}_0)$ is the target domain data distribution. It has the same marginal distribution $q(\boldsymbol{x}_t|\boldsymbol{x}_0, \hat{\boldsymbol{x}}_0)$ as in Eq. (2) for any $t \in [0, T]$ with the coefficients satisfying (proof in Appendix A.2)

$$f(t) = \frac{d \log \alpha_t(1 - \eta_t)}{dt}, \quad h(t) = \frac{\alpha_t}{1 - \eta_t}\frac{d\eta_t}{dt}, \quad g(t) = \sqrt{\frac{d\sigma_t^2}{dt} - 2\frac{d \log \alpha_t(1 - \eta_t)}{dt}\sigma_t^2}. \tag{6}$$

**Reverse Process.** The reverse of a diffusion process is also a diffusion process [2] which can similarly be described by a reverse-time SDE (proof in Appendix A.3):

$$d\boldsymbol{x}_t = \left[f(t)\boldsymbol{x}_t + h(t)\hat{\boldsymbol{x}}_0 - g^2(t)\nabla_{\boldsymbol{x}} \log q_t(\boldsymbol{x}_t)\right]dt + g(t)d\overline{\boldsymbol{w}}_t \tag{7}$$

where $\overline{\boldsymbol{w}}_t$ is also a standard Wiener process when time flows backwards. In this paper, we refer to this SDE as *Domain Shift* SDE (DoS-SDE).

**Score Matching.** The only unknown term in Eq. (7) is the *score function* $\nabla_{\boldsymbol{x}} \log q_t(\boldsymbol{x}_t)$ that can be estimated by training a score-based model on samples with score matching [39]. In practice, we use a neural network $\boldsymbol{\epsilon_\theta}(\boldsymbol{x}_t, \hat{\boldsymbol{x}}_0, t)$ conditioned on $\hat{\boldsymbol{x}}_0$, parameterized by $\boldsymbol{\theta}$, to estimate the scaled score function (alternatively referred to as noise), following [16, 40]. The parameter $\boldsymbol{\theta}$ is optimized by minimizing the following objectives:

$$\begin{aligned}
\boldsymbol{\theta}^* &= \arg\min_{\boldsymbol{\theta}} \mathbf{E}_t\left\{w(t)\mathbf{E}_{q_t(\boldsymbol{x}_t)}\left[||\boldsymbol{\epsilon_\theta}(\boldsymbol{x}_t, \hat{\boldsymbol{x}}_0, t) + \sigma_t\nabla_{\boldsymbol{x}} \log q_t(\boldsymbol{x}_t)||\right]\right\} \\
&= \arg\min_{\boldsymbol{\theta}} \mathbf{E}_t\left\{w(t)\mathbf{E}_{q_0(\boldsymbol{x}_0)}\mathbf{E}_{q(\boldsymbol{\epsilon})}\left[||\boldsymbol{\epsilon_\theta}(\boldsymbol{x}_t, \hat{\boldsymbol{x}}_0, t) - \boldsymbol{\epsilon}||\right]\right\},
\end{aligned} \tag{8}$$

where $w(t)$ is a weighting function, $\boldsymbol{x}_t = \alpha_t(\eta_t\hat{\boldsymbol{x}}_0 + (1 - \eta_t)\boldsymbol{x}_0) + \sigma_t\boldsymbol{\epsilon}$, and $\boldsymbol{\epsilon} \sim \mathcal{N}(\mathbf{0}, \boldsymbol{I})$.

Thus, we have completed the expression of the diffusion model using SDEs. Sampling from diffusion models can alternatively be seen as solving the corresponding diffusion DoS-SDEs.

## 3.3 Solvers for Diffusion DoS-SDEs

In this section, we present an exact formulation of the solution of diffusion DoS-SDEs and design efficient samplers for fast sampling. To facilitate the solution of equation Eq. (7), we utilize the data prediction model $\boldsymbol{x}_\theta(\boldsymbol{x}_t, \hat{\boldsymbol{x}}_0, t)$, which directly estimates the original target data $\boldsymbol{x}_0$ from the noisy samples. The relationship between score function and data prediction model is as follows (proof in Appendix A.4):

$$\nabla_{\boldsymbol{x}} \log q_t(\boldsymbol{x}_t) = -\frac{\boldsymbol{x}_t - (\alpha_t(1 - \eta_t)\boldsymbol{x}_\theta(\boldsymbol{x}_t, \hat{\boldsymbol{x}}_0, t) + \alpha_t \eta_t \hat{\boldsymbol{x}}_0)}{\sigma_t^2}. \tag{9}$$

In practice, we employ our trained noise prediction model $\boldsymbol{\epsilon}_\theta(\boldsymbol{x}_t, \hat{\boldsymbol{x}}_0, t)$ for data prediction $\boldsymbol{x}_\theta(\boldsymbol{x}_t, \hat{\boldsymbol{x}}_0, t)$ as described in Appendix A.4. By substituting Eq. (6) and Eq. (9) into Eq. (7) and introducing the substitutions $\lambda_t = \frac{\sigma_t}{\alpha_t(1-\eta_t)}$ and $\boldsymbol{y}_t = \frac{\boldsymbol{x}_t}{\alpha_t(1-\eta_t)}$ along with the notation $d\boldsymbol{w}_{\lambda_t} := \sqrt{\frac{d\lambda_t}{dt}} d\overline{\boldsymbol{w}}_t$, $\boldsymbol{x}_\lambda := \boldsymbol{x}_{t(\lambda)}$, $\boldsymbol{w}_\lambda := \boldsymbol{w}_{\lambda_t}$, we rewrite Eq. (7) w.r.t $\lambda$ as

$$d\boldsymbol{y}_\lambda = \frac{2}{\lambda} \boldsymbol{y}_\lambda d\lambda + \left[ \frac{1}{(1-\eta_\lambda)^2} d\eta_\lambda - \frac{\eta_\lambda}{1-\eta_\lambda} \frac{2}{\lambda} d\lambda \right] \hat{\boldsymbol{x}}_0 - \frac{2}{\lambda} \boldsymbol{x}_\theta(\boldsymbol{x}_\lambda, \hat{\boldsymbol{x}}_0, \lambda) d\lambda + \sqrt{2\lambda} d\boldsymbol{w}_\lambda \tag{10}$$

We propose the exact solution for Eq. (10) using the *variation-of-constants* formula, following [31, 9].

**Proposition 3.1** (Exact solution of diffusion DoS-SDEs). *Given an initial value $\boldsymbol{x}_s$ at time $s > 0$, the solution $\boldsymbol{x}_t$ for the diffusion DoS-SDEs defined in Eq. (7) at time $t \in [0, s]$ is as follows:*

$$\begin{aligned}
\boldsymbol{x}_t = &\frac{\alpha_t(1-\eta_t)}{\alpha_s(1-\eta_s)} \frac{\lambda_t^2}{\lambda_s^2} \boldsymbol{x}_s + \alpha_t(1-\eta_t)(\frac{\eta_t}{1-\eta_t} - \frac{\eta_s}{1-\eta_s} \frac{\lambda_t^2}{\lambda_s^2}) \hat{\boldsymbol{x}}_0 \\
&- \alpha_t(1-\eta_t) \int_{\lambda_s}^{\lambda_t} \frac{2\lambda_t^2}{\lambda^3} \boldsymbol{x}_\theta(\boldsymbol{x}_\lambda, \hat{\boldsymbol{x}}_0, \lambda) d\lambda + \alpha_t(1-\eta_t) \sqrt{\lambda_t^2 - \frac{\lambda_t^4}{\lambda_s^2}} \boldsymbol{z}_s,
\end{aligned} \tag{11}$$

*where $\lambda_t = \frac{\sigma_t}{\alpha_t(1-\eta_t)}$ and $\boldsymbol{z}_s \sim \mathcal{N}(\boldsymbol{0}, \boldsymbol{I})$.*

The detailed derivation of this proposition is provided in Appendix A.5. Notably, the nonlinear term in Eq. (11) involves the integration of a non-analytical neural network $\boldsymbol{x}_\theta(\boldsymbol{x}_\lambda, \hat{\boldsymbol{x}}_0, \lambda)$, which can be challenging to compute. For practical applicability, we employ Itô-Taylor expansion to approximate the integral of $\boldsymbol{x}_\theta$ from $\lambda_s$ to $\lambda_t$ to compute $\tilde{\boldsymbol{x}}_t$, thereby approximating $\boldsymbol{x}_t$. Additionally, we approximate the derivatives of $\boldsymbol{x}_\theta$ using the *forward differential method*. These approximations allow us to derive SDE solvers of any order for diffusion DoS-SDEs. For the sake of brevity, we employ a first-order solver for demonstration. In this case, Eq. (11) becomes

$$\begin{aligned}
\tilde{\boldsymbol{x}}_t = &\frac{\alpha_t(1-\eta_t)}{\alpha_s(1-\eta_s)} \frac{\lambda_t^2}{\lambda_s^2} \boldsymbol{x}_s + \underbrace{\alpha_t(1-\eta_t)(\frac{\eta_t}{1-\eta_t} - \frac{\eta_s}{1-\eta_s} \frac{\lambda_t^2}{\lambda_s^2}) \hat{\boldsymbol{x}}_0}_{\text{Domain Shift Guidance(DoSG)}} \\
&+ \alpha_t(1-\eta_t)(1 - \frac{\lambda_t^2}{\lambda_s^2}) \boldsymbol{x}_\theta(\boldsymbol{x}_s, \hat{\boldsymbol{x}}_0, s) + \alpha_t(1-\eta_t) \sqrt{\lambda_t^2 - \frac{\lambda_t^4}{\lambda_s^2}} \boldsymbol{z}_s.
\end{aligned} \tag{12}$$

The detailed derivation, as well as high-order solvers, are provided in Appendix A.6, and detailed algorithms are proposed in Appendix B. Typically, higher-order solvers converge even faster because of more accurate estimation of the the nonlinear integral term. The solvers provided for sampling allow us to iteratively generate HR images using a trained diffusion model. It is worth noting that Eq. (12) comprises four terms, including the additional linear term $\hat{\boldsymbol{x}}_0$, as compared to the ancestral sampling algorithm [16]. We refer to this additional term as the ***domain shift guidance*** (DoSG) which leverages prior information from the source domain and enhances the efficiency of inference.

## 4 Experiments

### 4.1 Experimental setup

For training, we train our DoSSR using a variety of datasets including DIV2K [1], DIV8K [15], Flickr2K [43], and OST [47]. To synthesize LR and HR training pairs, we adopt the degradation

Table 1: Quantitative comparison with state-of-the-art methods on both synthetic and real-world benchmarks, as well as comparison of latency and number of model parameters. NFE represents the number of function evaluations in the inference of diffusion models. The best and second best results of each metric are highlighted in **red** and blue, respectively.

| Datasets | Metrics | BSRGAN [54] | Real-ESRGAN [46] | LDL [23] | DASR [24] | StableSR [45] | ResShift [53] | DiffBIR [27] | SeeSR [51] | DoSSR |
|---|---|---|---|---|---|---|---|---|---|---|
| *DIV2k-Val* | PSNR ↑ | 24.58 | 24.29 | 23.83 | 24.47 | 23.36 | **24.65** | 23.67 | 23.68 | 23.98 |
| | SSIM ↑ | 0.6241 | **0.6338** | 0.6312 | 0.6277 | 0.5654 | 0.6148 | 0.5592 | 0.5987 | 0.6073 |
| | LPIPS ↓ | 0.3351 | **0.3112** | 0.3256 | 0.3543 | 0.3114 | 0.3349 | 0.3516 | 0.3195 | 0.3371 |
| | CLIPIQA ↑ | 0.5246 | 0.5276 | 0.5179 | 0.5036 | 0.6771 | 0.6065 | 0.6693 | 0.6935 | **0.7014** |
| | MUSIQ ↑ | 61.19 | 61.06 | 60.04 | 55.19 | 65.92 | 61.07 | 65.78 | **68.68** | 66.54 |
| | MANIQA ↑ | 0.3547 | 0.3795 | 0.3736 | 0.3165 | 0.4193 | 0.4107 | 0.4568 | 0.5041 | **0.5294** |
| | TOPIQ ↑ | 0.5456 | 0.5294 | 0.5142 | 0.4530 | 0.5974 | 0.5383 | 0.6142 | **0.6854** | 0.6766 |
| *RealSR* | PSNR ↑ | 26.38 | 25.69 | 25.28 | **27.02** | 24.65 | 26.26 | 24.81 | 25.14 | 24.18 |
| | SSIM ↑ | 0.7655 | 0.7615 | 0.7565 | **0.7714** | 0.7060 | 0.7404 | 0.6571 | 0.7194 | 0.6839 |
| | LPIPS ↓ | **0.2656** | 0.2709 | 0.2750 | 0.3134 | 0.3002 | 0.3469 | 0.3607 | 0.3007 | 0.3374 |
| | CLIPIQA ↑ | 0.5114 | 0.4485 | 0.4556 | 0.3198 | 0.6234 | 0.5473 | 0.6448 | 0.6699 | **0.7025** |
| | MUSIQ ↑ | 63.28 | 60.37 | 60.93 | 41.21 | 65.88 | 58.47 | 64.94 | **69.82** | 69.42 |
| | MANIQA ↑ | 0.3764 | 0.3733 | 0.3792 | 0.2461 | 0.4260 | 0.3836 | 0.4539 | 0.5406 | **0.5781** |
| | TOPIQ ↑ | 0.5502 | 0.5147 | 0.5124 | 0.3207 | 0.5743 | 0.4883 | 0.5722 | 0.6887 | **0.6985** |
| *DRealSR* | PSNR ↑ | 28.74 | 28.62 | 28.17 | **29.72** | 28.03 | 28.42 | 26.67 | 27.89 | 26.82 |
| | SSIM ↑ | 0.8033 | 0.8050 | 0.8126 | **0.8264** | 0.7523 | 0.7629 | 0.6548 | 0.7565 | 0.7298 |
| | LPIPS ↓ | 0.2858 | 0.2818 | **0.2792** | 0.3099 | 0.3284 | 0.4036 | 0.4517 | 0.3273 | 0.3689 |
| | CLIPIQA ↑ | 0.5091 | 0.4507 | 0.4473 | 0.3813 | 0.6357 | 0.5286 | 0.6391 | 0.6708 | **0.6776** |
| | MUSIQ ↑ | 57.16 | 54.28 | 53.95 | 42.41 | 58.51 | 49.73 | 60.91 | **65.09** | 64.40 |
| | MANIQA ↑ | 0.3424 | 0.3436 | 0.3444 | 0.2845 | 0.3867 | 0.3322 | 0.4486 | 0.5115 | **0.5214** |
| | TOPIQ ↑ | 0.5058 | 0.4621 | 0.4518 | 0.3482 | 0.5320 | 0.4380 | 0.5819 | 0.6574 | **0.6618** |
| *Real200* | CLIPIQA ↑ | 0.5910 | 0.5554 | 0.5508 | 0.5157 | 0.7272 | 0.6759 | 0.7170 | 0.7167 | **0.7437** |
| | MUSIQ ↑ | 67.65 | 66.12 | 65.80 | 61.26 | 70.63 | 66.98 | 68.92 | **72.14** | 71.62 |
| | MANIQA ↑ | 0.3882 | 0.3861 | 0.3921 | 0.3196 | 0.4838 | 0.4713 | 0.4869 | 0.5588 | **0.5794** |
| | TOPIQ ↑ | 0.5966 | 0.5530 | 0.5478 | 0.4793 | 0.6517 | 0.6124 | 0.6235 | 0.7142 | **0.7176** |
| NFE ↓ | | - | - | - | - | 200 | 15 | 50 | 50 | **5** |
| # Parameters | | 16.70M | 16.70M | 16.70M | 8.07M | 1409.1M | 173.9M | 1716.7M | 2283.7M | 1716.6M |
| Latency/Image ↓ | | 0.06s | 0.08s | 0.08s | 0.04s | 18.90s | 1.12s | 5.85s | 7.24s | 1.03s |

pipeline from RealESRGAN [46]. For the network architecture, we employ the LAControlNet [27] with SD 2.1-base[3] as the pretrained T2I model. In cases where LR images are severely degraded, potentially leading to the diffusion model mistaking degradation for semantic content, we implement RealESRNet [46] as a preprocessing step. This ensures our source domain consists of preprocessed LR images, thereby refining the input quality for better model training and performance. The model is fine-tuned for 50k iterations using the Adam optimizer [20], with a batch size of 32 and a learning rate set to $5 \times 10^{-5}$, on $512 \times 512$ resolution images.

For testing, we evaluate our method on both synthetic and real-world datasets, employing the same configuration as StableSR[4]. For synthetic data, we randomly crop 3K patches with a resolution of $512 \times 512$ from the DIV2K validation set [1], and degrade them following the degradation pipeline of RealESRGAN [46]. For real-world datasets, we generate LR images with a resolution of $128 \times 128$ by center-cropping on RealSR [4], DRealSR [50] and RealLR200 [51].

## 4.2 Comparisons with State-of-the-Arts

We compare DoSSR with the state-of-the-art real-world SR methods, including BSRGAN [54], Real-ESRGAN [46], LDL [23], DASR [24], StableSR [45], ResShift [53], DiffBIR [27], and SeeSR [51]. We use the publicly available codes and pretrained models to facilitate fair comparisons.

**Quantitative Comparison.** We show the quantitative comparison on the four synthetic and real-world datasets in Table 1. To comprehensively evaluate the performance of various methods, we utilize the following metrics[5] for quantitative comparison: reference-based metrics PSNR, SSIM [49], LPIPS [55], and non-reference metrics CLIPIQA [44], MUSIQ [19], MANIQA [52], TOPIQ [5].

---

[3]https://huggingface.co/stabilityai/stable-diffusion-2-1-base

[4]https://huggingface.co/datasets/Iceclear/StableSR-TestSets

[5]We use the repository available at https://github.com/chaofengc/IQA-PyTorch

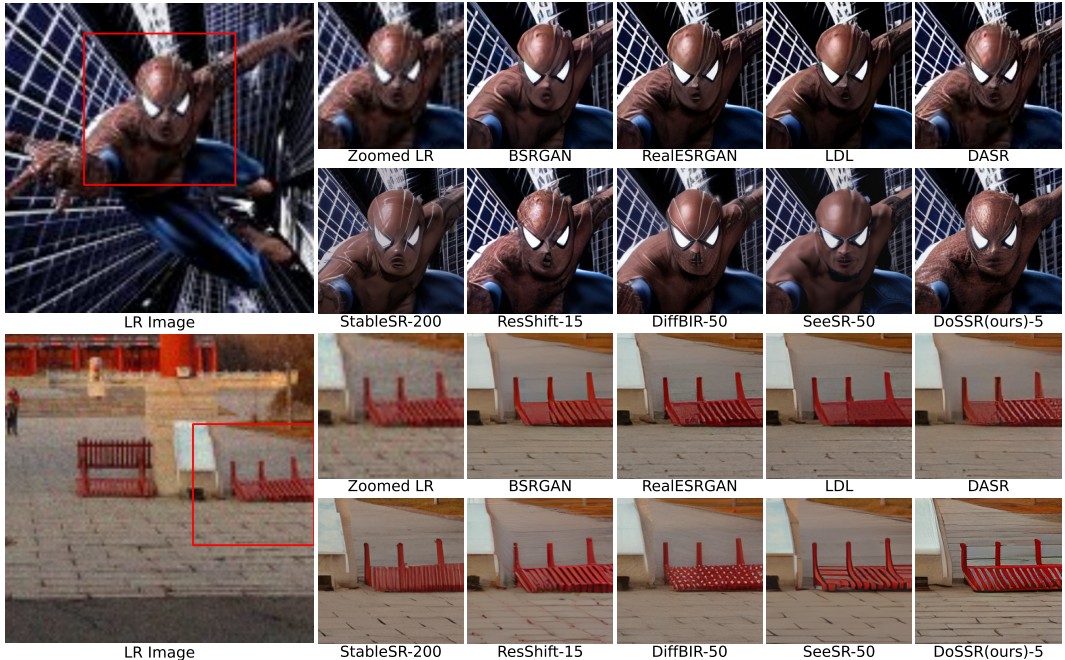

Figure 3: Qualitative comparisons of different steps of our DoSSR and other diffusion-based SR methods. The "-N" suffix denotes inference steps. Please zoom in for a better view.

Notably, DoSSR consistently achieves the highest scores in CLIPIQA, MANIQA, and TOPIQ, with the exception of being second in TOPIQ on DIV2K, and attains the second highest score in MUSIQ across all four datasets. At the same time, we also note that diffusion-based methods generally achieve poorer performance in reference metrics compared to GAN-based methods due to their ability to generate more realistic details at the expense of fidelity. Additionally, our DoSSR manages to achieve improved no-reference metric performance compared to the data presented in Table 1 as NFE increases slightly, a detail further elaborated on in Section 4.3.

**Qualitative Comparison.** Figs. 1(b), 3 present visual comparisons on real-world images. By leveraging learning of domain shift and introducing DoSG, our DoSSR efficiently generates high-quality texture details consistent with contents of the LR image. In the example of Fig. 1(b), GAN-based methods fail to faithfully reconstruct the grid texture of clothing, leading to notable degradation. StableSR and ResShift produce specific erroneous textures. Both SeeSR and ours successfully restore correct textures, while our results display clearer textures. Similarly, in the first example of Fig. 3, our DoSSR generates a more perceptually convincing Spider-Man face as well as textures, while in the second example, it produces more realistic and high-quality details of ground-laid bricks compared to other methods. More visual examples are provided in Fig. 7.

**Efficiency Comparison.** The comparative analysis of model parameters and latency for competing SR models is shown in Fig. 1(a) and Table 1. The latency is calculated on the ×4 SR task for 128×128 LR images with V100 GPU. StableSR, DiffBIR, SeeSR, and our DoSSR utilize the pretrained SD model, resulting in a similar parameter count, with SeeSR incorporating a prompt extractor to enhance SR results, making it the largest among these methods. ResShift, utilizing the network structure from LDM [35], is trained from scratch and has significantly fewer parameters. It employs a 15-step process to achieve faster inference speeds. Among the pretrained SD-based methods, DoSSR demonstrates superior performance efficiency, requiring only 5 function evaluations to achieve speeds 5-7 times faster than previous SD-based models such as SeeSR. Additionally, DoSSR not only demonstrates faster or comparable latency to ResShift but also achieves significantly better super-resolution performance.

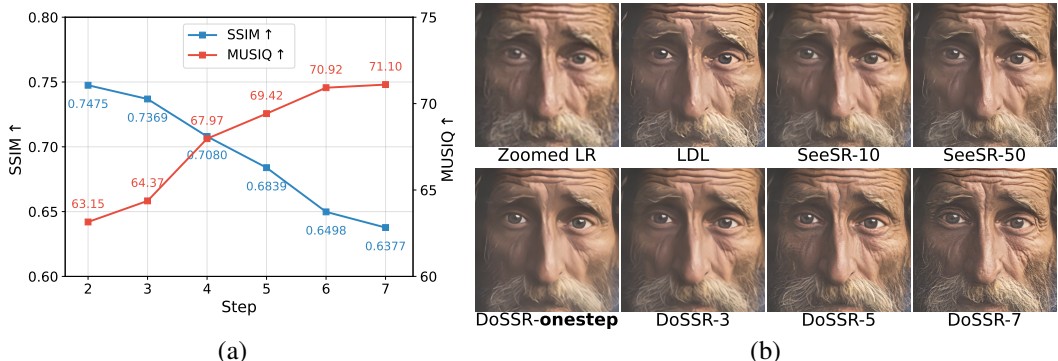

(a)

(b)

Figure 4: (a) Quality metrics vs. steps on *RealSR* Dataset. (b) Qualitative comparisons of different steps of our DoSSR with other methods. The suffix "-N" appended to the method name indicates the number of inference steps. Please zoom in for a better view.

## 4.3 Ablation Study

**Effectiveness of DoSG.** To verify the effectiveness of the DoSG introduced in the diffusion equation, we conduct an experiment using identical network architectures but with two different diffusion equations: the original diffusion equation as described by Ho *et al.* [16] and our newly formulated equation (Eq. (2)). To isolate the impact of DoSG from our shifting sequence design, we set $t_1 = T$, ensuring that the starting point of our inference in both scenarios approximates Gaussian noise. Quantitative comparisons can be found in the first two rows of Table 2. It is evident that the introduction of DoSG leads to a significant improvement across all metrics in the table, highlighting the effectiveness of DoSG in enhancing the performance of diffusion-based SR models. Additionally, it is worth noting that the original diffusion equation can be considered a special case within our framework where $\eta_t = 0$ and $t_1 = T$. Therefore, our sampler can accommodate the original diffusion equation, and for a fair comparison, we employ the same sampler for both models. More comprehensive comparison is provided in Appendix Table 6, where it can be seen that our DoSSR demonstrates superior performance compared to the corresponding order solver with DDPM, benefiting from the inclusion of DoSG in our DoS SDE-Solver.

**The selection of $t_1$.** The strating point $t_1$ serves as a pivotal parameter in DoSSR. We explore several options on the value of $t_1$ and show the corresponding final SR performance in Table 2. It can be observed that SR performance improves as $t_1$ gradually decreases from $T$ to $T/2$. However, further decreasing $t_1$ from $T/2$ to $3/T$ conversely compromises SR performance. Intuitively, a larger $t_1$ means less LR prior is preserved due to a larger magnitude of added noises, and the model behaves more like the vanilla pretrained model by hallucinating plausible HR contents; In contrast, a smaller $t_1$

| Method | | CLIPIQA↑ | MUSIQ↑ | MANIQA↑ | TOPIQ↑ |
|---|---|---|---|---|---|
| DDPM [16] | | 0.5379 | 54.09 | 0.3932 | 0.5180 |
| Domain Shift Diffusion-$t_1$ | $T$ | 0.5776 | 55.69 | 0.4181 | 0.5427 |
| | $2T/3$ | 0.6337 | 59.30 | 0.4589 | 0.5987 |
| | $\checkmark T/2$ | **0.6776** | **64.40** | **0.5214** | **0.6618** |
| | $T/3$ | 0.6490 | 61.76 | 0.4895 | 0.6260 |

Table 2: Comparison across various selections of starting point $t_1$, evaluated on the *DRealSR* dataset. The baseline method is DDPM, which employs the original diffusion equation. In all setups, inference is carried out over 5 steps.

means less noises, so the prediction is prone to be more consistent with the LR image, but without HR details. Hence, we set $t_1 = T/2$ by default for a good trade-off.

**The number of step.** We assess the impact of different inference steps on DoSSR by analyzing changes in representative metrics for both reference-based and non-reference-based evaluations, as shown in Fig. 4(a). As the number of inference steps increases, reference-based metrics tend to decline, suggesting a loss in fidelity, while non-reference metrics improve, indicating enhanced realism and detail in the generated images. We also conduct visual comparisons in Fig. 4(b). Our DoSSR achieves performance comparable to SeeSR in just 5 steps and produces more realistic details in 7 steps. Remarkably, DoSSR is capable of delivering satisfactory results ***even with just a***

*single step*, achieving 0.5115 MANIQA score and 0.6258 CLIPIQA score on the *RealSR* dataset, significantly boosting the efficiency of diffusion-based methods. More visual examples are provided in Fig. 9, where it can be observed that increasing the number of steps yields more realistic details.

**The order of our sampler.** We provide a suite of solvers for sampling in our DoSSR model, including a first-order solver presented in Eq. (11), and more advanced second- and third-order solvers detailed in Appendix A.6. We investigate the impact of samplers with different orders on our experimental results through qualitative and quantitative comparisons, as illustrated in Table 3 and Fig. 10. From Table 3, it becomes evident that high-order samplers can achieve superior non-reference metrics under the same limited inference step conditions. This is because the acceleration of

| Order | CLIPIQA↑ | MUSIQ↑ | MANIQA↑ | TOPIQ↑ |
|-------|----------|--------|---------|--------|
| 1 | 0.5907 | 59.12 | 0.4686 | 0.5907 |
| 2 | 0.6749 | 64.09 | 0.5196 | 0.6571 |
| 3 | **0.6776** | **64.40** | **0.5214** | **0.6618** |

Table 3: Comparison of performance of different sampler orders on the *DRealSR* dataset. In all setups, inference is carried out over 5 steps.

higher-order samplers allows diffusion models to generate more details, as demonstrated in the first example of Fig. 10, where the tower generated by the high-order sampler exhibits richer textures. More comprehensive comparison is provided in Appendix Table 6. In our implementation, we use third-order sampler by default.

## 5 Conclusion

In this paper, we present DoSSR, a diffusion-based super-resolution framework that significantly enhances both efficiency and performance by integrating a domain shift strategy with pretrained diffusion models. This approach not only enhances generative capacity but also enhances further inference efficiency through our novel proposed DoS-SDEs formulation and customized solvers. Empirical validation on diverse SR benchmarks confirms that DoSSR achieves a 5-7 times speed improvement over existing methods, setting a new state-of-the-art. Our work paves the way for more efficient diffusion-based solutions in image super-resolution.

**Limitation.** Despite the strong overall performance demonstrated by the proposed DoSSR, it occasionally generates visually unfriendly details when employing an unfavorable random seed, a challenge also encountered by other diffusion-based methods. Typically, we fix the random seed for all image super-resolution tasks to stabilize the results, but this particular seed may not be suitable for certain specific images. As depicted in Fig. 8, different initializations of random seeds result in significant variations in the details of the lion's eyes. Some of the initialized random seeds produce eyes that are reasonable and acceptable, while others exhibit noticeable inconsistencies with LR. For bad cases, we can also obtain a satisfactory result by adjusting the random seed multiple times. However, this often requires numerous attempts, and the quality of the results heavily relies on luck. This inspires us to find a suitable initialization for each specific LR image, which can enhance the performance of the model. Hence, for diffusion-based methods, exploring how to obtain a reasonable random seed based on known LR images may be a future research direction.

**Societal impact.** Our advancements in the diffusion-based image super-resolution model, DoSSR, present both positive and negative societal impacts. On the positive side, it enhances medical imaging, potentially leading to more accurate diagnoses and reducing the need for invasive procedures. In surveillance, it aids in better identification and tracking, improving public safety. Moreover, in remote sensing and environmental monitoring, it facilitates informed decision-making for disaster management and environmental conservation. However, there are concerns regarding privacy and surveillance. Enhanced resolution capabilities could infringe upon privacy rights and lead to increased surveillance in public spaces, raising questions about civil liberties. Additionally, in digital media, while high-resolution imagery enhances visual content, it may perpetuate unrealistic beauty standards and digital manipulation, impacting self-esteem. In summary, while DoSSR brings promising advancements, it's crucial to address concerns around privacy, security, and digital ethics to ensure responsible and ethical deployment of the technology.

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

# A Mathematical Details

## A.1 Derivation of Eq.(3)

According to Eq. (2), we can express $\boldsymbol{x}_t$ as a linear combination of $\boldsymbol{x}_0, \hat{\boldsymbol{x}}_0$ and a noise variable $\boldsymbol{\epsilon}$:

$$\boldsymbol{x}_t = \alpha_t(\eta_t \hat{\boldsymbol{x}}_0 + (1 - \eta_t)\boldsymbol{x}_0) + \sigma_t \boldsymbol{\epsilon}, \text{ where } \boldsymbol{\epsilon} \sim \mathcal{N}(\boldsymbol{0}, \boldsymbol{I}). \tag{13}$$

Subsequently, the relationship between $\boldsymbol{x}_t$ and $\boldsymbol{x}_{t-1}$ is derived as follows:

$$\boldsymbol{x}_t = \alpha_t(\eta_{t-1}\hat{\boldsymbol{x}}_0 + (1 - \eta_{t-1})\boldsymbol{x}_0 + (\eta_t - \eta_{t-1})(\hat{\boldsymbol{x}}_0 - \boldsymbol{x}_0)) + \sigma_t \boldsymbol{\epsilon}$$

$$= \frac{\alpha_t}{\alpha_{t-1}}\left[\alpha_{t-1}(\eta_{t-1}\hat{\boldsymbol{x}}_0 + (1 - \eta_{t-1})\boldsymbol{x}_0) + \sigma_{t-1}\boldsymbol{\epsilon}_1\right] + \alpha_t(\eta_t - \eta_{t-1})\boldsymbol{e}_0 + \sqrt{\sigma_t^2 - \frac{\alpha_t^2}{\alpha_{t-1}^2}\sigma_{t-1}^2}\boldsymbol{\epsilon}_2$$

$$= \frac{\alpha_t}{\alpha_{t-1}}\boldsymbol{x}_{t-1} + \alpha_t(\eta_t - \eta_{t-1})\boldsymbol{e}_0 + \sqrt{\sigma_t^2 - \frac{\alpha_t^2}{\alpha_{t-1}^2}\sigma_{t-1}^2}\boldsymbol{\epsilon}_2$$

where $\boldsymbol{e}_0 = \hat{\boldsymbol{x}}_0 - \boldsymbol{x}_0$, and $\boldsymbol{\epsilon}, \boldsymbol{\epsilon}_1, \boldsymbol{\epsilon}_2 \sim \mathcal{N}(\boldsymbol{0}, \boldsymbol{I})$. Taking into account $\alpha_t^2 + \sigma_t^2 = 1$, the above equation can be further simplified as follows:

$$\boldsymbol{x}_t = \frac{\alpha_t}{\alpha_{t-1}}\boldsymbol{x}_{t-1} + \alpha_t(\eta_t - \eta_{t-1})\boldsymbol{e}_0 + \sqrt{1 - \frac{\alpha_t^2}{\alpha_{t-1}^2}}\boldsymbol{\epsilon}_2, \tag{14}$$

Hence, the transition distribution between $\boldsymbol{x}_t$ and $\boldsymbol{x}_{t-1}$ is as follows:

$$q(\boldsymbol{x}_t|\boldsymbol{x}_{t-1}, \hat{\boldsymbol{x}}_0) = \mathcal{N}(\boldsymbol{x}_t; \frac{\alpha_t}{\alpha_{t-1}}\boldsymbol{x}_{t-1} + \alpha_t(\eta_t - \eta_{t-1})\boldsymbol{e}_0, 1 - \frac{\alpha_t^2}{\alpha_{t-1}^2}\boldsymbol{I}), \ t = 1, 2, \cdots, T, \tag{15}$$

## A.2 Derivation of Eq.(6)

In this section, we derive the coefficients of the forward SDE. Discretizing Eq. (5) yields:

$$\boldsymbol{x}_{t+\Delta t} - \boldsymbol{x}_t = f(t)\boldsymbol{x}_t\Delta t + h(t)\hat{\boldsymbol{x}}_0\Delta t + g(t)\sqrt{\Delta t}\boldsymbol{z}_1, \text{ where } \boldsymbol{z}_1 \sim \mathcal{N}(\boldsymbol{0}, \boldsymbol{I})$$

$$\boldsymbol{x}_{t+\Delta t} = (f(t)\Delta t + 1)\boldsymbol{x}_t + h(t)\hat{\boldsymbol{x}}_0\Delta t + g(t)\sqrt{\Delta t}\boldsymbol{z}_1. \tag{16}$$

Substituting Eq. (13) into Eq. (16), we have

$$\boldsymbol{x}_{t+\Delta t} = (f(t)\Delta t + 1)[\alpha_t(\eta_t\hat{\boldsymbol{x}}_0 + (1 - \eta_t)\boldsymbol{x}_0) + \sigma_t\boldsymbol{z}_2] + h(t)\hat{\boldsymbol{x}}_0\Delta t + g(t)\sqrt{\Delta t}\boldsymbol{z}_1$$
$$= \alpha_t(f(t)\Delta t + 1)(1 - \eta_t)\boldsymbol{x}_0 + [\alpha_t\eta_t(f(t)\Delta t + 1) + h(t)\Delta t]\hat{\boldsymbol{x}}_0 \tag{17}$$
$$+ \sqrt{(f(t)\Delta t + 1)^2\sigma_t^2 + g(t)^2}\tilde{\boldsymbol{z}},$$

where $\boldsymbol{z}_2, \tilde{\boldsymbol{z}} \sim \mathcal{N}(\boldsymbol{0}, \boldsymbol{I})$. For Eq. (13) at time $t + \Delta t$, we have:

$$\boldsymbol{x}_{t+\Delta t} = \alpha_{t+\Delta t}(\eta_{t+\Delta t}\hat{\boldsymbol{x}}_0 + (1 - \eta_{t+\Delta t})\boldsymbol{x}_0) + \sigma_{t+\Delta t}\boldsymbol{\epsilon}. \tag{18}$$

Equating the corresponding parts of Eq. (17) and Eq. (18) yields:

$$\begin{cases} \alpha_{t+\Delta t}(1 - \eta_{t+\Delta t}) = \alpha_t(f(t)\Delta t + 1)(1 - \eta_t) \\ \alpha_{t+\Delta t}\eta_{t+\Delta t} = \alpha_t\eta_t(f(t)\Delta t + 1) + h(t)\Delta t \\ \sigma_{t+\Delta t}^2 = [f(t)\Delta t + 1]^2\sigma_t^2 + g(t)^2\Delta t \end{cases} \tag{19}$$

Then, letting $\Delta t \to 0$, the aforementioned three equations can be solved separately to yield:

$$\begin{cases} f(t) = \dfrac{d\log\alpha_t(1 - \eta_t)}{dt} \\ h(t) = \dfrac{\alpha_t}{1 - \eta_t}\dfrac{d\eta_t}{dt} \\ g(t) = \sqrt{\dfrac{d\sigma_t^2}{dt} - 2\dfrac{d\log\alpha_t(1 - \eta_t)}{dt}\sigma_t^2} \end{cases} \tag{20}$$

## A.3 Derivation of Eq.(7)

As outlined in Sec. 3.2, the forward process can be expressed as the SDE shown in Eq. (5). In accordance with the Fokker-Plank Equation [34], we obtain:

$$\frac{\partial q_t(\boldsymbol{x}_t)}{\partial t} = -\nabla_{\boldsymbol{x}}\{[f(t)\boldsymbol{x}_t + h(t)\hat{\boldsymbol{x}}_0]q_t(\boldsymbol{x}_t)\} + \frac{\partial}{\partial \boldsymbol{x}_i \partial \boldsymbol{x}_j}[\frac{1}{2}g^2(t)q_t(\boldsymbol{x}_t)]$$

$$= -\nabla_{\boldsymbol{x}}\{[f(t)\boldsymbol{x}_t + h(t)\hat{\boldsymbol{x}}_0]q_t(\boldsymbol{x}_t)\} + \nabla_{\boldsymbol{x}}[\frac{1}{2}g^2(t)\nabla_{\boldsymbol{x}}q_t(\boldsymbol{x}_t)]$$

$$= -\nabla_{\boldsymbol{x}}\{[f(t)\boldsymbol{x}_t + h(t)\hat{\boldsymbol{x}}_0]q_t(\boldsymbol{x}_t)\} + \nabla_{\boldsymbol{x}}[\frac{1}{2}g^2(t)q_t(\boldsymbol{x}_t)\nabla_{\boldsymbol{x}}\log q_t(\boldsymbol{x}_t)]$$

$$= -\nabla_{\boldsymbol{x}}\{[f(t)\boldsymbol{x}_t + h(t)\hat{\boldsymbol{x}}_0 - \frac{1}{2}g^2(t)\nabla_{\boldsymbol{x}}\log q_t(\boldsymbol{x}_t)]q_t(\boldsymbol{x}_t)\},$$

where $q(\boldsymbol{x}_t)$ denotes the probability density function of state $\boldsymbol{x}_t$. Most process defined by a forward-time or conventional diffusion equation model possess a corresponding reverse-time model [2], which can be formulated as:

$$d\boldsymbol{x}_t = \mu(t, \boldsymbol{x}_t)dt + \sigma(t, \boldsymbol{x}_t)d\overline{\boldsymbol{w}}_t \tag{21}$$

According to the backward Fokker-Plank Equation [34], we have:

$$\frac{\partial q_t(\boldsymbol{x}_t)}{\partial t} = -\nabla_{\boldsymbol{x}}[\mu(t, \boldsymbol{x}_t)q_t(\boldsymbol{x}_t)] - \frac{\partial}{\partial \boldsymbol{x}_i \partial \boldsymbol{x}_j}[\frac{1}{2}\sigma^2(t, \boldsymbol{x}_t)q_t(\boldsymbol{x}_t)]$$

$$= -\nabla_{\boldsymbol{x}}[\mu(t, \boldsymbol{x}_t)q_t(\boldsymbol{x}_t)] - \nabla_{\boldsymbol{x}}[\frac{1}{2}\sigma^2(t, \boldsymbol{x}_t)\nabla_{\boldsymbol{x}}q_t(\boldsymbol{x}_t)]$$

$$= -\nabla_{\boldsymbol{x}}[\mu(t, \boldsymbol{x}_t)q_t(\boldsymbol{x}_t)] - \nabla_{\boldsymbol{x}}[\frac{1}{2}\sigma^2(t, \boldsymbol{x}_t)q_t(\boldsymbol{x}_t)\nabla_{\boldsymbol{x}}\log q_t(\boldsymbol{x}_t)]$$

$$= -\nabla_{\boldsymbol{x}}\{[\mu(t, \boldsymbol{x}_t) + \frac{1}{2}\sigma^2(t, \boldsymbol{x}_t)\nabla_{\boldsymbol{x}}\log q_t(\boldsymbol{x}_t)]q_t(\boldsymbol{x}_t)\}.$$

Our goal is for the reverse process to have the same distribution as the forward process, specifically:

$$\mu(t, \boldsymbol{x}_t) + \frac{1}{2}\sigma^2(t, \boldsymbol{x}_t)\nabla_{\boldsymbol{x}}\log q_t(\boldsymbol{x}_t) = f(t)\boldsymbol{x}_t + h(t)\hat{\boldsymbol{x}}_0 - \frac{1}{2}g^2(t)\nabla_{\boldsymbol{x}}\log q_t(\boldsymbol{x}_t). \tag{22}$$

Typically, we set $\sigma(t, \boldsymbol{x}_t) = g(t)$ [40], yielding:

$$\mu(t, \boldsymbol{x}_t) = f(t)\boldsymbol{x}_t + h(t)\hat{\boldsymbol{x}}_0 - g^2(t)\nabla_{\boldsymbol{x}}\log q_t(\boldsymbol{x}_t). \tag{23}$$

Therefore, the reverse-time SDE can be expressed as follows:

$$d\boldsymbol{x}_t = \Big[f(t)\boldsymbol{x}_t + h(t)\hat{\boldsymbol{x}}_0 - g^2(t)\nabla_{\boldsymbol{x}}\log q_t(\boldsymbol{x}_t)\Big]dt + g(t)d\overline{\boldsymbol{w}}_t. \tag{24}$$

## A.4 Derivation of Eq.(9)

The data prediction model $\boldsymbol{x}_\theta(\boldsymbol{x}_t, \hat{\boldsymbol{x}}_0, t)$ directly estimates the original target data $\boldsymbol{x}_0$ from the noisy samples, indicating that $\boldsymbol{x}_\theta(\boldsymbol{x}_t, \hat{\boldsymbol{x}}_0, t) \approx \boldsymbol{x}_0$. Based on Eq. (2), the expression for $q_t(\boldsymbol{x}_t)$ can be formulated as follows:

$$q_t(\boldsymbol{x}_t) = \frac{1}{\sqrt{2\pi}\sigma_t}\exp(-\frac{[\boldsymbol{x}_t - (\alpha_t(1-\eta_t)\boldsymbol{x}_0 + \alpha_t\eta_t\hat{\boldsymbol{x}}_0)]^2}{2\sigma_t^2}). \tag{25}$$

Hence, score function is:

$$\nabla_{\boldsymbol{x}}\log q_t(\boldsymbol{x}_t) = -\frac{\boldsymbol{x}_t - (\alpha_t(1-\eta_t)\boldsymbol{x}_0 + \alpha_t\eta_t\hat{\boldsymbol{x}}_0)}{\sigma_t^2}. \tag{26}$$

Substituting $\boldsymbol{x}_\theta(\boldsymbol{x}_t, \hat{\boldsymbol{x}}_0, t) \approx \boldsymbol{x}_0$, we can establish the relationship between the score function and the data prediction model:

$$\nabla_{\boldsymbol{x}}\log q_t(\boldsymbol{x}_t) = -\frac{\boldsymbol{x}_t - (\alpha_t(1-\eta_t)\boldsymbol{x}_\theta(\boldsymbol{x}_t, \hat{\boldsymbol{x}}_0, t) + \alpha_t\eta_t\hat{\boldsymbol{x}}_0)}{\sigma_t^2} \tag{27}$$

Furthermore, Eq. (8) shows that the noise prediction model is to estimate

$$\boldsymbol{\epsilon_\theta}(\boldsymbol{x}_t, \hat{\boldsymbol{x}}_0, t) \approx -\sigma_t\nabla_{\boldsymbol{x}}\log q_t(\boldsymbol{x}_t). \tag{28}$$

Hence, the relationship between the noise prediction model and the data prediction model is:

$$\boldsymbol{\epsilon_\theta}(\boldsymbol{x}_t, \hat{\boldsymbol{x}}_0, t) = \frac{\boldsymbol{x}_t - (\alpha_t(1-\eta_t)\boldsymbol{x}_\theta(\boldsymbol{x}_t, \hat{\boldsymbol{x}}_0, t) + \alpha_t\eta_t\hat{\boldsymbol{x}}_0)}{\sigma_t}, \tag{29}$$

indicating that we easily use trained noise prediction model for data prediction through the equation.

## A.5 Proof of Proposition 3.1

In this section, we derive the solution to the equation Eq. (7). By substituting Eq. (9) and Eq. (20) into Eq. (7), we obtain:

$$
d\boldsymbol{x}_t = \left\{ \frac{1}{\alpha_t(1-\eta_t)} \frac{d[\alpha_t(1-\eta_t)]}{dt} \boldsymbol{x}_t + \frac{\alpha_t}{1-\eta_t} \frac{d\eta_t}{dt} \hat{\boldsymbol{x}}_0 \right.
$$
$$
\left. - \left[ 2\sigma_t \frac{d\sigma_t}{dt} - 2\sigma_t^2 \frac{d[\alpha_t(1-\eta_t)]}{dt} \right] - \frac{\boldsymbol{x}_t - (\alpha_t(1-\eta_t)\boldsymbol{x}_\theta(\boldsymbol{x}_t,\hat{\boldsymbol{x}}_0,t) + \alpha_t\eta_t\hat{\boldsymbol{x}}_0)}{\sigma_t^2} \right\} dt \qquad (30)
$$
$$
+ \sqrt{\frac{d\sigma_t^2}{dt} - 2\frac{d\log\alpha_t(1-\eta_t)}{dt} \sigma_t^2} \, d\overline{\boldsymbol{w}}_t.
$$

Combining like terms, we get:

$$
d\boldsymbol{x}_t = \left[ \frac{2}{\sigma_t} \frac{d\sigma_t}{dt} - \frac{1}{\alpha_t(1-\eta_t)} \frac{d[\alpha_t(1-\eta_t)]}{dt} \right] \boldsymbol{x}_t
$$
$$
+ \left\{ \frac{\alpha_t}{1-\eta_t} \frac{d\eta_t}{dt} - \alpha_t\eta_t \left[ \frac{2}{\sigma_t} \frac{d\sigma_t}{dt} - \frac{2}{\alpha_t(1-\eta_t)} \frac{d[\alpha_t(1-\eta_t)]}{dt} \right] \right\} \hat{\boldsymbol{x}}_0
$$
$$
- 2\alpha_t(1-\eta_t) \left[ \frac{1}{\sigma_t} \frac{d\sigma_t}{dt} - \frac{1}{\alpha_t(1-\eta_t)} \frac{d[\alpha_t(1-\eta_t)]}{dt} \right] \boldsymbol{x}_\theta(\boldsymbol{x}_t,\hat{\boldsymbol{x}}_0,t) \qquad (31)
$$
$$
+ \sqrt{\frac{d\sigma_t^2}{dt} - 2\frac{d\log\alpha_t(1-\eta_t)}{dt} \sigma_t^2} \, d\overline{\boldsymbol{w}}_t.
$$

Subsequently, dividing both sides by $\alpha_t(1-\eta_t)$ simultaneously, we have:

$$
\frac{1}{\alpha_t(1-\eta_t)} d\boldsymbol{x}_t = \left[ \frac{2}{\sigma_t} \frac{d\sigma_t}{dt} - \frac{1}{\alpha_t(1-\eta_t)} \frac{d[\alpha_t(1-\eta_t)]}{dt} \right] \frac{\boldsymbol{x}_t}{\alpha_t(1-\eta_t)}
$$
$$
+ \left\{ \frac{1}{(1-\eta_t)^2} \frac{d\eta_t}{dt} - \frac{\eta_t}{(1-\eta_t)} \left[ \frac{2}{\sigma_t} \frac{d\sigma_t}{dt} - \frac{2}{\alpha_t(1-\eta_t)} \frac{d[\alpha_t(1-\eta_t)]}{dt} \right] \right\} \hat{\boldsymbol{x}}_0
$$
$$
- 2 \left[ \frac{1}{\sigma_t} \frac{d\sigma_t}{dt} - \frac{1}{\alpha_t(1-\eta_t)} \frac{d[\alpha_t(1-\eta_t)]}{dt} \right] \boldsymbol{x}_\theta(\boldsymbol{x}_t,\hat{\boldsymbol{x}}_0,t) \qquad (32)
$$
$$
+ \sqrt{\frac{2\sigma_t}{[\alpha_t(1-\eta_t)]^2} \frac{d\sigma_t}{dt} - \frac{2\sigma_t^2}{[\alpha_t(1-\eta_t)]^3} \frac{d[\alpha_t(1-\eta_t)]}{dt}} \, d\overline{\boldsymbol{w}}_t.
$$

Let $\lambda_t = \frac{\sigma_t}{\alpha_t(1-\eta_t)}$. Then $\lambda_t$ is monotonically increasing, and we have:

$$
\frac{d\lambda_t}{dt} = \frac{1}{\alpha_t(1-\eta_t)} \frac{d\sigma_t}{dt} - \frac{\sigma_t}{[\alpha_t(1-\eta_t)]^2} \frac{d[\alpha_t(1-\eta_t)]}{dt}. \qquad (33)
$$

Therefore, we have:

$$
\frac{1}{\sigma_t} \frac{d\sigma_t}{dt} - \frac{1}{\alpha_t(1-\eta_t)} \frac{d[\alpha_t(1-\eta_t)]}{dt} = \frac{1}{\lambda_t} \frac{d\lambda_t}{dt}. \qquad (34)
$$

By performing the variable substitution $\boldsymbol{y}_t = \frac{\boldsymbol{x}_t}{\alpha_t(1-\eta_t)}$ and then substituting Eq. (34) into Eq. (32), we can simplify to obtain:

$$
d\boldsymbol{y}_t = \left\{ \frac{2}{\lambda_t} \frac{d\lambda_t}{dt} \boldsymbol{y}_t + \left[ \frac{1}{(1-\eta_t)^2} \frac{d\eta_t}{dt} - \frac{\eta_t}{1-\eta_t} \left( \frac{2}{\lambda_t} \frac{d\lambda_t}{dt} \right) \right] \hat{\boldsymbol{x}}_0 - \frac{2}{\lambda_t} \frac{d\lambda_t}{dt} \boldsymbol{x}_\theta(\boldsymbol{x}_t,\hat{\boldsymbol{x}}_0,t) \right\} dt + \sqrt{2\lambda_t \frac{d\lambda_t}{dt}} \, d\overline{\boldsymbol{w}}_t.
$$
$$
(35)
$$

Denoting $d\boldsymbol{w}_{\lambda_t} := \sqrt{\frac{d\lambda_t}{dt}} \, d\overline{\boldsymbol{w}}_t$, $\boldsymbol{x}_\lambda := \boldsymbol{x}_{t(\lambda)}$, $\boldsymbol{w}_\lambda := \boldsymbol{w}_{\lambda_t}$, we rewrite the equation above w.r.t $\lambda$ as

$$
d\boldsymbol{y}_\lambda = \frac{2}{\lambda} \boldsymbol{y}_\lambda d\lambda + \left[ \frac{1}{(1-\eta_\lambda)^2} d\eta_\lambda - \frac{\eta_\lambda}{1-\eta_\lambda} \frac{2}{\lambda} d\lambda \right] \hat{\boldsymbol{x}}_0 - \frac{2}{\lambda} \boldsymbol{x}_\theta(\boldsymbol{x}_\lambda,\hat{\boldsymbol{x}}_0,\lambda) d\lambda + \sqrt{2\lambda} d\boldsymbol{w}_\lambda. \qquad (36)
$$

Utilizing the *variation-of-constants* formula to solve the equation above, we obtain

$$
\boldsymbol{y}_t = e^{\int_{\lambda_s}^{\lambda_t} \frac{2}{\lambda} d\lambda} \boldsymbol{y}_s + \int_{\lambda_s}^{\lambda_t} e^{\int_{\lambda}^{\lambda_t} \frac{2}{\tau} d\tau} \left[ \frac{1}{(1-\eta_\lambda)^2} d\eta_\lambda - \frac{\eta_\lambda}{1-\eta_\lambda} \frac{2}{\lambda} d\lambda \right] \hat{\boldsymbol{x}}_0
$$
$$
- \int_{\lambda_s}^{\lambda_t} e^{\int_{\lambda}^{\lambda_t} \frac{2}{\tau} d\tau} \frac{2}{\lambda} \boldsymbol{x}_\theta(\boldsymbol{x}_\lambda,\hat{\boldsymbol{x}}_0,\lambda) d\lambda + \int_{\lambda_s}^{\lambda_t} e^{\int_{\lambda}^{\lambda_t} \frac{2}{\tau} d\tau} \sqrt{2\lambda} d\boldsymbol{w}_\lambda. \qquad (37)
$$

Simplifying and substituting back $\boldsymbol{x}_t = \alpha_t(1-\eta_t)\boldsymbol{y}_t$, we obtain

$$
\begin{aligned}
\boldsymbol{x}_t ={}& \frac{\alpha_t(1-\eta_t)}{\alpha_s(1-\eta_s)}\frac{\lambda_t^2}{\lambda_s^2}\boldsymbol{x}_s + \alpha_t(1-\eta_t)\left(\frac{\eta_t}{1-\eta_t} - \frac{\eta_s}{1-\eta_s}\frac{\lambda_t^2}{\lambda_s^2}\right)\hat{\boldsymbol{x}}_0 \\
& - \alpha_t(1-\eta_t)\int_{\lambda_s}^{\lambda_t}\frac{2\lambda_t^2}{\lambda^3}\boldsymbol{x}_\theta(\boldsymbol{x}_\lambda, \hat{\boldsymbol{x}}_0, \lambda)d\lambda + \alpha_t(1-\eta_t)\sqrt{\lambda_t^2 - \frac{\lambda_t^4}{\lambda_s^2}}\boldsymbol{z}_s.
\end{aligned}
\tag{38}
$$

Thus, we obtain the exact solution to the DoS-SDEs.

## A.6 Derivation of Solvers for Diffusion DoS-SDEs

Denote $\boldsymbol{x}_\theta^{(n)}(\boldsymbol{x}_\lambda, \hat{\boldsymbol{x}}_0, \lambda) := \frac{d^n\boldsymbol{x}_\theta(\boldsymbol{x}_\lambda, \hat{\boldsymbol{x}}_0, \lambda)}{d\lambda^n}$ as the $n$-th order total derivative of $\boldsymbol{x}_\theta(\boldsymbol{x}_\lambda, \lambda)$ w.r.t $\lambda$. For $k \geq 1$, the $k-1$-th order Itô-Taylor expansion of $\boldsymbol{x}_\theta(\boldsymbol{x}_\lambda, \hat{\boldsymbol{x}}_0, \lambda)$ w.r.t $\lambda$ at $s$ is

$$
\boldsymbol{x}_\theta(\boldsymbol{x}_\lambda, \hat{\boldsymbol{x}}_0, \lambda) = \sum_{n=0}^{k-1}\frac{(\lambda - \lambda_s)^n}{n!}\boldsymbol{x}_\theta^{(n)}(\boldsymbol{x}_s, \hat{\boldsymbol{x}}_0, s) + \mathcal{R}_k,
\tag{39}
$$

where the residual $\mathcal{R}_k$ comprises of deterministic iterated integrals of length greater than $k$ and all iterated with at least one stochastic component.

Substituting the above Itô-Taylor expansion into Eq. (38) yields

$$
\begin{aligned}
\boldsymbol{x}_t ={}& \frac{\alpha_t(1-\eta_t)}{\alpha_s(1-\eta_s)}\frac{\lambda_t^2}{\lambda_s^2}\boldsymbol{x}_s + \alpha_t(1-\eta_t)\left(\frac{\eta_t}{1-\eta_t} - \frac{\eta_s}{1-\eta_s}\frac{\lambda_t^2}{\lambda_s^2}\right)\hat{\boldsymbol{x}}_0 \\
& - \alpha_t(1-\eta_t)\sum_{n=0}^{k-1}\boldsymbol{x}_\theta^{(n)}(\boldsymbol{x}_s, \hat{\boldsymbol{x}}_0, s)\int_{\lambda_s}^{\lambda_t}\frac{2\lambda_t^2}{\lambda^3}\frac{(\lambda - \lambda_s)^n}{n!}d\lambda + \alpha_t(1-\eta_t)\sqrt{\lambda_t^2 - \frac{\lambda_t^4}{\lambda_s^2}}\boldsymbol{z}_s + \tilde{\mathcal{R}}_k,
\end{aligned}
\tag{40}
$$

where $\tilde{\mathcal{R}}_k$ can be easily obtained from $\mathcal{R}_k$ and the integral $\int_{\lambda_s}^{\lambda_t}\frac{2\lambda_t^2}{\lambda^3}\frac{(\lambda - \lambda_s)^n}{n!}d\lambda$ can be analytically computed by repeated applying $n$ times of integration-by-parts. By dropping the $\mathcal{R}_k$ error and approximating the first $k-1$-th total derivatives with *forward differential method*, we can derive $k$-th order SDE solvers for diffusion DoS-SDEs. In fact, it is inaccurate to call it "order" when $k \geq 2$, because the proposed algorithm has a global error of at least $\mathcal{O}(\lambda - \lambda_s)$ [14]. Thus, only when $k = 1$, it is referred to as a first-order solver with a strong convergence guarantee, as stated in [14]. Nevertheless, for practical convenience, we still refer to this approximation as $k$-th order. Here we present the expressions for first-order as well as second and third-order solvers. We name such solvers as *DoS-SDE Solver* overall, and *DoS-SDE Solver-k* for a specific order $k$.

**DoS-SDE Solver-1**   When $k = 1$, the integral becomes

$$
-\int_{\lambda_s}^{\lambda_t}\frac{2\lambda_t^2}{\lambda^3}\boldsymbol{x}_\theta(\boldsymbol{x}_\lambda, \lambda)d\lambda \approx -\lambda_t^2\int_{\lambda_s}^{\lambda_t}\frac{2}{\lambda^3}d\lambda\boldsymbol{x}_\theta(\boldsymbol{x}_s, \hat{\boldsymbol{x}}_0, s) = \left(1 - \frac{\lambda_t^2}{\lambda_s^2}\right)\boldsymbol{x}_\theta(\boldsymbol{x}_s, \hat{\boldsymbol{x}}_0, s).
\tag{41}
$$

Substituting into Eq.(38), we obtain first-order solver for DoS-SDEs

$$
\begin{aligned}
\boldsymbol{x}_t ={}& \frac{\alpha_t(1-\eta_t)}{\alpha_s(1-\eta_s)}\frac{\lambda_t^2}{\lambda_s^2}\boldsymbol{x}_s + \alpha_t(1-\eta_t)\left(\frac{\eta_t}{1-\eta_t} - \frac{\eta_s}{1-\eta_s}\frac{\lambda_t^2}{\lambda_s^2}\right)\hat{\boldsymbol{x}}_0 \\
& + \alpha_t(1-\eta_t)\left(1 - \frac{\lambda_t^2}{\lambda_s^2}\right)\boldsymbol{x}_\theta(\boldsymbol{x}_s, \hat{\boldsymbol{x}}_0, s) + \alpha_t(1-\eta_t)\sqrt{\lambda_t^2 - \frac{\lambda_t^4}{\lambda_s^2}}\boldsymbol{z}_s.
\end{aligned}
\tag{42}
$$

**DoS-SDE Solver-2**   When $k = 2$, the integral in Eq.(40) becomes

$$
\begin{aligned}
& -\sum_{n=0}^{1}\boldsymbol{x}_\theta^{(n)}(\boldsymbol{x}_s, \hat{\boldsymbol{x}}_0, s)\int_{\lambda_s}^{\lambda_t}\frac{2\lambda_t^2}{\lambda^3}\frac{(\lambda - \lambda_s)^n}{n!}d\lambda \\
& = -\int_{\lambda_s}^{\lambda_t}\frac{2\lambda_t^2}{\lambda^3}d\lambda\boldsymbol{x}_\theta(\boldsymbol{x}_s, \hat{\boldsymbol{x}}_0, s) - \int_{\lambda_s}^{\lambda_t}\frac{2\lambda_t^2}{\lambda^3}(\lambda - \lambda_s)d\lambda\boldsymbol{x}_\theta^{(1)}(\boldsymbol{x}_s, \hat{\boldsymbol{x}}_0, s) \\
& = \left(1 - \frac{\lambda_t^2}{\lambda_s^2}\right)\boldsymbol{x}_\theta(\boldsymbol{x}_s, \hat{\boldsymbol{x}}_0, s) - \frac{(\lambda_t - \lambda_s)^2}{\lambda_s}\boldsymbol{x}_\theta^{(1)}(\boldsymbol{x}_s, \hat{\boldsymbol{x}}_0, s)
\end{aligned}
\tag{43}
$$

Substituting into Eq.(40), we obtain 2-th order solver for DoS-SDEs

$$
\begin{aligned}
\boldsymbol{x}_t &= \frac{\alpha_t(1-\eta_t)}{\alpha_s(1-\eta_s)}\frac{\lambda_t^2}{\lambda_s^2}\boldsymbol{x}_s + \alpha_t(1-\eta_t)(\frac{\eta_t}{1-\eta_t}-\frac{\eta_s}{1-\eta_s}\frac{\lambda_t^2}{\lambda_s^2})\hat{\boldsymbol{x}}_0 + \alpha_t(1-\eta_t)\sqrt{\lambda_t^2-\frac{\lambda_t^4}{\lambda_s^2}}\boldsymbol{z}_s \\
&\quad + \alpha_t(1-\eta_t)(1-\frac{\lambda_t^2}{\lambda_s^2})\boldsymbol{x}_\theta(\boldsymbol{x}_s,\hat{\boldsymbol{x}}_0,s) - \alpha_t(1-\eta_t)\frac{(\lambda_t-\lambda_s)^2}{\lambda_s}\boldsymbol{x}_\theta^{(1)}(\boldsymbol{x}_s,\hat{\boldsymbol{x}}_0,s),
\end{aligned} \tag{44}
$$

where $\boldsymbol{x}_\theta^{(1)}(\boldsymbol{x}_s,\hat{\boldsymbol{x}}_0,s)$ can be estimated by *forward differential method*. We have

$$
\boldsymbol{x}_\theta^{(1)}(\boldsymbol{x}_s,\hat{\boldsymbol{x}}_0,s) \approx \frac{\boldsymbol{x}_\theta(\boldsymbol{x}_s,\hat{\boldsymbol{x}}_0,s) - \boldsymbol{x}_\theta(\boldsymbol{x}_r,\hat{\boldsymbol{x}}_0,r)}{\lambda_s - \lambda_r}, \tag{45}
$$

where time $t < s < r$ and $\boldsymbol{x}_\theta(\boldsymbol{x}_r,\hat{\boldsymbol{x}}_0,r)$ represents the output of the network at the previous time step.

**DoS-SDE Solver-3**   Samely, when $k=3$, the integral in Eq.(40) becomes

$$
\begin{aligned}
&-\sum_{n=0}^{2}\boldsymbol{x}_\theta^{(n)}(\boldsymbol{x}_s,\hat{\boldsymbol{x}}_0,s)\int_{\lambda_s}^{\lambda_t}\frac{2\lambda_t^2}{\lambda^3}\frac{(\lambda-\lambda_s)^n}{n!}d\lambda \\
&= (1-\frac{\lambda_t^2}{\lambda_s^2})\boldsymbol{x}_\theta(\boldsymbol{x}_s,\hat{\boldsymbol{x}}_0,s) - \frac{(\lambda_t-\lambda_s)^2}{\lambda_s}\boldsymbol{x}_\theta^{(1)}(\boldsymbol{x}_s,\hat{\boldsymbol{x}}_0,s) - \int_{\lambda_s}^{\lambda_t}\frac{\lambda_t^2}{\lambda^3}(\lambda-\lambda_s)^2 d\lambda\, \boldsymbol{x}_\theta^{(2)}(\boldsymbol{x}_s,\hat{\boldsymbol{x}}_0,s) \\
&= (1-\frac{\lambda_t^2}{\lambda_s^2})\boldsymbol{x}_\theta(\boldsymbol{x}_s,\hat{\boldsymbol{x}}_0,s) - \frac{(\lambda_t-\lambda_s)^2}{\lambda_s}\boldsymbol{x}_\theta^{(1)}(\boldsymbol{x}_s,\hat{\boldsymbol{x}}_0,s) \\
&\quad + \Big[\frac{(\lambda_s-3\lambda_t)(\lambda_s-\lambda_t)}{2} - \lambda_t^2 ln(\frac{\lambda_t}{\lambda_s})\Big]\boldsymbol{x}_\theta^{(2)}(\boldsymbol{x}_s,\hat{\boldsymbol{x}}_0,s)
\end{aligned} \tag{46}
$$

Substituting into Eq.(40), we obtain 3-th order solver for DoS-SDEs

$$
\begin{aligned}
\boldsymbol{x}_t &= \frac{\alpha_t(1-\eta_t)}{\alpha_s(1-\eta_s)}\frac{\lambda_t^2}{\lambda_s^2}\boldsymbol{x}_s + \alpha_t(1-\eta_t)(\frac{\eta_t}{1-\eta_t}-\frac{\eta_s}{1-\eta_s}\frac{\lambda_t^2}{\lambda_s^2})\hat{\boldsymbol{x}}_0 + \alpha_t(1-\eta_t)\sqrt{\lambda_t^2-\frac{\lambda_t^4}{\lambda_s^2}}\boldsymbol{z}_s \\
&\quad + \alpha_t(1-\eta_t)(1-\frac{\lambda_t^2}{\lambda_s^2})\boldsymbol{x}_\theta(\boldsymbol{x}_s,\hat{\boldsymbol{x}}_0,s) - \alpha_t(1-\eta_t)\frac{(\lambda_t-\lambda_s)^2}{\lambda_s}\boldsymbol{x}_\theta^{(1)}(\boldsymbol{x}_s,\hat{\boldsymbol{x}}_0,s) \\
&\quad + \alpha_t(1-\eta_t)\Big[\frac{(\lambda_s-3\lambda_t)(\lambda_s-\lambda_t)}{2} - \lambda_t^2 ln(\frac{\lambda_t}{\lambda_s})\Big]\boldsymbol{x}_\theta^{(2)}(\boldsymbol{x}_s,\hat{\boldsymbol{x}}_0,s)
\end{aligned} \tag{47}
$$

where $\boldsymbol{x}_\theta^{(1)}(\boldsymbol{x}_s,\hat{\boldsymbol{x}}_0,s)$ and $\boldsymbol{x}_\theta^{(2)}(\boldsymbol{x}_s,\hat{\boldsymbol{x}}_0,s)$ can be estimated by *forward differential method*. We have

$$
\boldsymbol{x}_\theta^{(1)}(\boldsymbol{x}_s,\hat{\boldsymbol{x}}_0,s) \approx \frac{\boldsymbol{x}_\theta(\boldsymbol{x}_s,\hat{\boldsymbol{x}}_0,s) - \boldsymbol{x}_\theta(\boldsymbol{x}_r,\hat{\boldsymbol{x}}_0,r)}{\lambda_s - \lambda_r}, \tag{48}
$$

where time $t < s < r$ and $\boldsymbol{x}_\theta(\boldsymbol{x}_r,\hat{\boldsymbol{x}}_0,r)$ represents the output of the network at the previous time step. And

$$
\boldsymbol{x}_\theta^{(2)}(\boldsymbol{x}_s,\hat{\boldsymbol{x}}_0,s) \approx \frac{\boldsymbol{x}_\theta^{(1)}(\boldsymbol{x}_s,\hat{\boldsymbol{x}}_0,s) - \boldsymbol{x}_\theta^{(1)}(\boldsymbol{x}_r,\hat{\boldsymbol{x}}_0,r)}{\frac{\lambda_s-\lambda_q}{2}}, \tag{49}
$$

where time $t < s < r < q$ and $\boldsymbol{x}_\theta^{(1)}(\boldsymbol{x}_s,\hat{\boldsymbol{x}}_0,s)$ and $\boldsymbol{x}_\theta^{(1)}(\boldsymbol{x}_r,\hat{\boldsymbol{x}}_0,r)$ respectively represent the approximations of the first-order derivatives at the current and previous steps.

Detailed algorithms for our solvers are proposed in Sec. B

## A.7   Comparative Analysis of DoSSR and ResShift

Previous work has introduced a method called ResShift [53], which shortens the length of the Markov chain in the diffusion process through residual shifting to achieve efficient super-resolution in diffusion models. In this section, we theoretically elaborate on the similarities and differences between our method and ResShift.

ResShift expresses the forward diffusion process in the form of residual shifting, as shown in the following equation:

$$q(\boldsymbol{x}_t|\boldsymbol{x}_0,\boldsymbol{y}_0) = \mathcal{N}(\boldsymbol{x}_t; \boldsymbol{x}_0 + \eta_t \boldsymbol{e}_0, k^2 \eta_t \boldsymbol{I}),\ t = 1, 2, \cdots, T, \tag{50}$$

where $\boldsymbol{e}_0 = \boldsymbol{y}_0 - \boldsymbol{x}_0$ respensents the residual between the LR image $\boldsymbol{y}_0$ and the HR image $\boldsymbol{x}_0$. Hence, it can be rewrite as,

$$q(\boldsymbol{x}_t|\boldsymbol{x}_0,\boldsymbol{y}_0) = \mathcal{N}(\boldsymbol{x}_t; \eta_t \boldsymbol{y}_0 + (1 - \eta_t)\boldsymbol{x}_0, k^2 \eta_t \boldsymbol{I}),\ t = 1, 2, \cdots, T. \tag{51}$$

Therefore, essentially, the ResShift concept also represents a linear combination of the source domain and the target domain. However, the crucial factor lies in our design of the diffusion equation, which determines whether we can *effectively utilize the diffusion prior*. This is our most significant differentiating point. Currently, the mainstream approach to leveraging diffusion prior involves fine-tuning large-scale pretrained diffusion models to achieve SR. As we all know, Stable Diffusion, a typical representative of large-scale diffusion models, employs the diffusion equation of DDPM [16]. Its forward process can be expressed as:

$$q(\boldsymbol{x}_t|\boldsymbol{x}_0) = \mathcal{N}(\boldsymbol{x}_t; \alpha_t \boldsymbol{x}_0, \sigma_t^2 \boldsymbol{I}),\ t = 1, 2, \cdots, T, \tag{52}$$

where $\alpha_t, \sigma_t \geq 0$ and $\alpha_t^2 + \sigma_t^2 = 1$, referred to as *noise schedule*. If we consider $\eta_t \boldsymbol{y}_0 + (1 - \eta_t)\boldsymbol{x}_0$ as a whole, we can intuitively observe that Eq. (51) lacks a decay coefficient $\alpha_t$ compared to Eq. (52). Therefore, applying Eq. (51) to fine-tune a pretrained Stable Diffusion model poses significant challenges. The equation proposed by our DoSSR is restated as follows:

$$q(\boldsymbol{x}_t|\boldsymbol{x}_0,\hat{\boldsymbol{x}}_0) = \mathcal{N}(\boldsymbol{x}_t; \alpha_t(\eta_t \hat{\boldsymbol{x}}_0 + (1 - \eta_t)\boldsymbol{x}_0), \sigma_t^2 \boldsymbol{I}),\ t = 1, 2, \cdots, T, \tag{53}$$

Our diffusion equation incorporates the noise schedule of Stable Diffusion, enabling seamless compatibility with existing DDPM-type diffusion models. Therefore, in this sense, our method is specifically designed as a diffusion equation tailored to adapt to Stable Diffusion.

In fact, Eq. (51) and Eq. (52) (or Eq. (53)) belong to two different types of diffusion models, which correspond to the discretizations of two types of SDEs(VE SDE and VP SDE) [40], respectively.

The diffusion equation that satisfies the discretizations of VE SDEs typically takes the following form:

$$q(\boldsymbol{x}_t|\boldsymbol{x}_0) = \mathcal{N}(\boldsymbol{x}_t; \boldsymbol{x}_0, \sigma_t^2 \boldsymbol{I}),\ t = 1, 2, \cdots, T, \tag{54}$$

where $\sigma_t$ increases with $t$, and $\sigma_T$ is typically a very large number, ensuring that $q(\boldsymbol{x}_T) = \mathcal{N}(\boldsymbol{x}_0; \sigma_T^2 \boldsymbol{I}) \approx \mathcal{N}(\boldsymbol{0}; \sigma_T^2 \boldsymbol{I})$. Therefore, Eq. (51) can be considered as a variant of the above equation. By setting hyperparameters such that $k^2 \eta_t = \sigma_t^2$, it appears possible to fine-tune a diffusion model pretrained with Eq. (54) as the diffusion equation, thereby leveraging the diffusion prior. However, this contradicts its original intention, as the starting point of inference almost approximates Gaussian noise, retaining little of the prior information of LR. Our design of shifting sequence $\{\eta_t\}_{t=1}^{T}$ is aimed at addressing this issue, which the ResShift lacks.

The second type of diffusion equation that satisfies the discretizations of VP SDEs is of the DDPM type. In Eq. (52), the noise schedule satisfies the noise schedule $0 \leq \sigma_t \leq 1$ and it is typically set to $\sigma_T \approx 1, \alpha_T \approx 0$ to ensure that $q(\boldsymbol{x}_T) = \mathcal{N}(\alpha_T \boldsymbol{x}_0; \sigma_T^2 \boldsymbol{I}) \approx \mathcal{N}(\boldsymbol{0}; \boldsymbol{I})$. To my knowledge, most large-scale pretrained diffusion models, represented by Stable Diffusion, adopt the VP-type (DDPM-type) diffusion equation. Therefore, our design is highly significant. Moreover, our theory is not limited to the analysis of discrete cases but extends to more general continuous cases, expressed as SDEs. Based on this, we have developed our fast samplers for our DoSSR. These are the distinctions between our work and ResShift.

To further demonstrate the effect of the diffusion prior, we conducted an additional comparative experiment with ResShift, as detailed in Appendix C.

# B Pseudocode

Here, algorithms for first, second, and third-order solvers for DoS-SDEs are presented as follows.

---

**Algorithm 1** DoSSR Solver-1.

---

**Require:** starting point $t_1$, used time steps $\{t_i\}_{i=1}^N$, nosie schedule $\alpha_t$ and $\sigma_t$, preprocessed LR image $\hat{x}_0$, data prediction model $x_\theta$.

1: $x_{t_1} \leftarrow \alpha_{t_1}\hat{x}_0 + \sigma_{t_1}\epsilon$        ▷ initial value
2: **for** $i \leftarrow 2$ to $N$ **do**
3:      $DoSG \leftarrow \alpha_{t_i}(1-\eta_{t_i})(\frac{\eta_{t_i}}{1-\eta_{t_i}} - \frac{\eta_{t_{i-1}}}{1-\eta_{t_{i-1}}}\frac{\lambda_{t_i}^2}{\lambda_{t_{i-1}}^2})\hat{x}_0$        ▷ *domain shift guidance*
4:      $Linear\ Term \leftarrow \frac{\alpha_{t_i}(1-\eta_{t_i})}{\alpha_{t_{i-1}}(1-\eta_{t_{i-1}})}\frac{\lambda_{t_i}^2}{\lambda_{t_{i-1}}^2}x_{t_{i-1}}$
5:      $Noise\ Term \leftarrow \alpha_{t_i}(1-\eta_{t_i})\sqrt{\lambda_{t_i}^2 - \frac{\lambda_{t_i}^4}{\lambda_{t_{i-1}}^2}}\hat{x}_0$
6:      $PAT \leftarrow \alpha_{t_i}(1-\eta_{t_i})(1 - \frac{\lambda_{t_i}^2}{\lambda_{t_{i-1}}^2})x_\theta(x_{t_{i-1}}, \hat{x}_0, t_{i-1})$        ▷ *Prediction Approximation Term*
7:      $x_{t_i} \leftarrow Linear\ Term + DoSG + PAT + Noise\ Term$
8: **end for**
9: **Return:** $x_{t_N}$

---

**Algorithm 2** DoSSR Solver-2.

---

**Require:** starting point $t_1$, used time steps $\{t_i\}_{i=1}^N$, noise schedule $\alpha_t$ and $\sigma_t$, preprocessed LR image $\hat{x}_0$, data prediction model $x_\theta$.

1: $x_{t_1} \leftarrow \alpha_{t_1}\hat{x}_0 + \sigma_{t_1}\epsilon$        ▷ initial value
2: $Q \leftarrow None$
3: **for** $i \leftarrow 2$ to $N$ **do**
4:      $DoSG \leftarrow \alpha_{t_i}(1-\eta_{t_i})(\frac{\eta_{t_i}}{1-\eta_{t_i}} - \frac{\eta_{t_{i-1}}}{1-\eta_{t_{i-1}}}\frac{\lambda_{t_i}^2}{\lambda_{t_{i-1}}^2})\hat{x}_0$        ▷ domain shift guidance
5:      $Linear\ Term \leftarrow \frac{\alpha_{t_i}(1-\eta_{t_i})}{\alpha_{t_{i-1}}(1-\eta_{t_{i-1}})}\frac{\lambda_{t_i}^2}{\lambda_{t_{i-1}}^2}x_{t_{i-1}}$
6:      $Noise\ Term \leftarrow \alpha_{t_i}(1-\eta_{t_i})\sqrt{\lambda_{t_i}^2 - \frac{\lambda_{t_i}^4}{\lambda_{t_{i-1}}^2}}\hat{x}_0$
7:      **if** $Q = None$ **then**
8:          $PAT \leftarrow \alpha_{t_i}(1-\eta_{t_i})(1 - \frac{\lambda_{t_i}^2}{\lambda_{t_{i-1}}^2})x_\theta(x_{t_{i-1}}, \hat{x}_0, t_{i-1})$
9:      **else**
10:          $\mathbf{D}_i \leftarrow \frac{x_\theta(x_{t_{i-1}}, \hat{x}_0, t_{i-1}) - x_\theta(x_{t_{i-2}}, \hat{x}_0, t_{i-2})}{\lambda_{t_{i-1}} - \lambda_{t_{i-2}}}$        ▷ first order derivative
11:          $PAT \leftarrow \alpha_{t_i}(1-\eta_{t_i})(1 - \frac{\lambda_{t_i}^2}{\lambda_{t_{i-1}}^2})x_\theta(x_{t_{i-1}}, \hat{x}_0, t_{i-1}) - \alpha_{t_i}(1-\eta_{t_i})\frac{(\lambda_{t_i} - \lambda_{t_{i-1}})^2}{\lambda_{t_{i-1}}}\mathbf{D}_i$
12:      **end if**
13:      $Q \leftarrow x_\theta(x_{t_{i-1}}, \hat{x}_0, t_{i-1})$        ▷ save last network output
14:      $x_{t_i} \leftarrow Linear\ Term + DoSG + PAT + Noise\ Term$
15: **end for**
16: **Return:** $x_{t_N}$

---

**Algorithm 3** DoSSR Solver-3.

---

**Require:** starting point $t_1$, used time steps $\{t_i\}_{i=1}^N$, noise schedule $\alpha_t$ and $\sigma_t$, preprocessed LR image $\hat{\boldsymbol{x}}_0$, data prediction model $\boldsymbol{x}_\theta$.

1: $\boldsymbol{x}_{t_1} \leftarrow \alpha_{t_1}\hat{\boldsymbol{x}}_0 + \sigma_{t_1}\boldsymbol{\epsilon}$          $\triangleright$ initial value

2: $Q \leftarrow None, Q_d \leftarrow None$

3: **for** $i \leftarrow 2$ to $N$ **do**

4:      $DoSG \leftarrow \alpha_{t_i}(1-\eta_{t_i})\big(\frac{\eta_{t_i}}{1-\eta_{t_i}} - \frac{\eta_{t_{i-1}}}{1-\eta_{t_{i-1}}}\frac{\lambda_{t_i}^2}{\lambda_{t_{i-1}}^2}\big)\hat{\boldsymbol{x}}_0$          $\triangleright$ domain shift guidance

5:      $Linear\ Term \leftarrow \frac{\alpha_{t_i}(1-\eta_{t_i})}{\alpha_{t_{i-1}}(1-\eta_{t_{i-1}})}\frac{\lambda_{t_i}^2}{\lambda_{t_{i-1}}^2}\boldsymbol{x}_{t_{i-1}}$

6:      $Noise\ Term \leftarrow \alpha_{t_i}(1-\eta_{t_i})\sqrt{\lambda_{t_i}^2 - \frac{\lambda_{t_i}^4}{\lambda_{t_{i-1}}^2}}\hat{\boldsymbol{x}}_0$

7:      **if** $Q = None$ and $Q_d = None$ **then**

8:          $PAT \leftarrow \alpha_{t_i}(1-\eta_{t_i})(1-\frac{\lambda_{t_i}^2}{\lambda_{t_{i-1}}^2})\boldsymbol{x}_\theta(\boldsymbol{x}_{t_{i-1}}, \hat{\boldsymbol{x}}_0, t_{i-1})$

9:      **else if** $Q \neq None$ and $Q_d = None$ **then**

10:          $\mathbf{D}_i \leftarrow \frac{\boldsymbol{x}_\theta(\boldsymbol{x}_{t_{i-1}},\hat{\boldsymbol{x}}_0,t_{i-1}) - \boldsymbol{x}_\theta(\boldsymbol{x}_{t_{i-2}},\hat{\boldsymbol{x}}_0,t_{i-2})}{\lambda_{t_{i-1}} - \lambda_{t_{i-2}}}$          $\triangleright$ first order derivative

11:          $PAT \leftarrow \alpha_{t_i}(1-\eta_{t_i})(1-\frac{\lambda_{t_i}^2}{\lambda_{t_{i-1}}^2})\boldsymbol{x}_\theta(\boldsymbol{x}_{t_{i-1}},\hat{\boldsymbol{x}}_0,t_{i-1}) - \alpha_{t_i}(1-\eta_{t_i})\frac{(\lambda_{t_i}-\lambda_{t_{i-1}})^2}{\lambda_{t_{i-1}}}\mathbf{D}_i$

12:          $Q_d \leftarrow D_i$

13:      **else**

14:          $\mathbf{D}_i \leftarrow \frac{\boldsymbol{x}_\theta(\boldsymbol{x}_{t_{i-1}},\hat{\boldsymbol{x}}_0,t_{i-1}) - \boldsymbol{x}_\theta(\boldsymbol{x}_{t_{i-2}},\hat{\boldsymbol{x}}_0,t_{i-2})}{\lambda_{t_{i-1}} - \lambda_{t_{i-2}}}$          $\triangleright$ first order derivative

15:          $\mathbf{U}_i \leftarrow \frac{\mathbf{D}_i - \mathbf{D}_{i-1}}{\frac{\lambda_{t_{i-1}} - \lambda_{t_{i-3}}}{2}}$          $\triangleright$ second order derivative

16:          $PAT \leftarrow \alpha_{t_i}(1-\eta_{t_i})(1-\frac{\lambda_{t_i}^2}{\lambda_{t_{i-1}}^2})\boldsymbol{x}_\theta(\boldsymbol{x}_{t_{i-1}},\hat{\boldsymbol{x}}_0,t_{i-1}) - \alpha_{t_i}(1-\eta_{t_i})\frac{(\lambda_{t_i}-\lambda_{t_{i-1}})^2}{\lambda_{t_{i-1}}}\mathbf{D}_i +$
$\alpha_{t_i}(1-\eta_{t_i})\Big[\frac{(\lambda_{t_{i-1}}-3\lambda_{t_i})(\lambda_{t_{i-1}}-\lambda_{t_i})}{2} - \lambda_{t_i}^2\ln(\frac{\lambda_{t_i}}{\lambda_{t_{i-1}}})\Big]\mathbf{U}_i$

17:          $Q_d \leftarrow D_i$

18:      **end if**

19:      $Q \leftarrow \boldsymbol{x}_\theta(\boldsymbol{x}_{t_{i-1}}, \hat{\boldsymbol{x}}_0, t_{i-1})$          $\triangleright$ save last network output

20:      $\boldsymbol{x}_{t_i} \leftarrow Linear\ Term + DoSG + PAT + Noise\ Term$

21: **end for**

22: **Return:** $\boldsymbol{x}_{t_N}$

---

## C    Additional Experiment Results

### C.1    Ablation on Diffusion Prior

To demonstrate the impact of the diffusion prior, we conduct a comparative experiment with the ResShift model [53], which does not leverage a pretrained diffusion model. Considering that ResShift is trained on ImageNet [11] while our DoSSR model is trained on commonly used datasets for super-resolution tasks (e.g., DIV2K [1]), to eliminate the influence of the training dataset, we retrain our DoSSR model using ImageNet as well. To highlight the effect of the diffusion prior, we only utilize a subset of ImageNet as our training data. This subset consists of randomly selected 10 images from each of the 1000 categories in ImageNet, totaling 10,000 images, as illustrated in Table 4. It can be observed that our DoSSR achieves superior metrics compared to ResShift despite utilizing significantly fewer training data and epochs for iteration. This indicates that leveraging the diffusion prior is highly beneficial for super-resolution tasks, even without utilizing commonly used high-resolution datasets for training, we can still achieve satisfactory results. We also perform qualitative comparisons between our retrained DoSSR and ResShift, as illustrated in Fig. 5. It is evident that the utilization of the diffusion prior significantly enhances the quality of the generated images, both in terms of fidelity and realism.

Table 4: Comparison of performance between our retrained DoSSR and ResShift models on the *DRealSR* dataset. For fair comparison, we employ our first-order sampler for inference, running it 15 times to match ResShift's default setting.

| Method | Diffusion Prior | Training Setup | | | Evalution Metrics | | | |
|--------|-----------------|----------------|--|--|-------------------|--|--|--|
| | | Training Dataset | Num of Iters | Traing Method | SSIM ↑ | LPIPS ↓ | MUSIQ ↑ | MANIQA ↑ |
| ResShift | ✗ | ImageNet(∼1280k) | 500k | Train from scratch | 0.7629 | 0.4036 | 49.73 | 0.3322 |
| DoSSR | ✓ | sub-ImageNet(10k) | 50k | Finetune | 0.7824 | 0.2943 | 53.94 | 0.3672 |

Table 5: Comparison of performance with other methods on the *RealSRSet* and *RealSR* datasets. NFE represents the number of function evaluations in the inference. * involves retraining using the same training data and identical network architecture as our model.

| Method | Training Dataset | RealSRSet | | RealSR | | NFE↓ |
|--------|------------------|-----------|--|--------|--|------|
| | | MUSIQ↑ | MANIQA↑ | MUSIQ↑ | MANIQA↑ | |
| ColdDiff* | DIV2K+DIV8K+Flickr2K+OST(∼ 15k) | 58.19 | 0.3194 | 47.42 | 0.2783 | 5 |
| ResShift* | DIV2K+DIV8K+Flickr2K+OST(∼ 15k) | 63.90 | 0.4505 | 56.01 | 0.4001 | 5 |
| DoSSR | DIV2K+DIV8K+Flickr2K+OST(∼ 15k) | **73.35** | **0.6169** | **69.42** | **0.5781** | 5 |
| FlowIE | ImageNet(∼ 1280k) | 61.63 | 0.3611 | 56.51 | 0.3284 | 1 |
| FlowIE* | DIV2K+DIV8K+Flickr2K+OST(∼ 15k) | 60.48 | 0.3644 | 50.82 | 0.3228 | 1 |
| DoSSR | DIV2K+DIV8K+Flickr2K+OST(∼ 15k) | **69.42** | **0.5554** | **62.69** | **0.5115** | 1 |

## C.2 Compare with other formulations of diffusion process

Aside from ResShift, we also compare our method with other plausible alternative formulations of diffusion processes, which encompass ColdDiff [3] and Rectified Flow [29]. ColdDiff is similar to DDPM, but differs by employing alternative degradation methods, such as blurring and masking, rather than additive Gaussian noise as used in DDPM. In our case for image super-resolusion, we select the blur degradation. Note that ColdDiff is originally applied in the pixel space, while we have to implement it in the latent space for fair comparison with our method. Rectified Flow defines the forward process as $x_t = t\hat{x}_0 + (1 - t)x_0$ where $\hat{x}_0$ is the data sample and $x_0$ is Gaussian noise. In our case for image super-resolusion, we choose FlowIE [59] as an implementation of rectified flow which replaces $x_0$ with the LR image as the starting point.

For a fair comparison, we reimplemented the three methods—ResShift [53], ColdDiff [3], and FlowIE [59]—using the same network architecture, training dataset, and initialization as our method. The results are shown in the Table 5. Results of ColdDiff is significantly worse than other methods. The main reason can be summerized as two aspects. First, applying bluring kernels to latent features is not equivalent to applying them to raw images, while ColdDiff is originally designed for the latter. Second, the blurring kernel can only be designed in a hand-craft manner and may not represent the real-world degradation, so when tested with real LR images there would be a domain gap. The limilation of hand-crafted degradation is also well-known in many previous image restoration works. In contrast, other methods, including our DoSSR, does not assume a fixed degradation, so the performance is much better. Because FlowIE emphasizes single step evaluation, we follow its setting and compare DoSSR with 1-setp evaluation with it. It can be see that FlowIE is also not satisfactory and largely underperforms our DoSSR ($-11.87\%$ MUSIQ). This is mainly attributed to FlowIE's bad adaptation from the DDPM pretrained T2I model, since the learning objective of flow matching differs from score matching. By contrast, our design makes full use of the DDPM pretrained weights so performs much better. ResShift also significantly underperforms our method, similar to FlowIE, because its formulation does not account for adaptation from the pretrained diffusion prior, as discussed in Appendix A.7 and C.1. To summarize, our formulation differs from other alternatives especially in terms of *better leverage and adaptation from DDPM pretrained T2I models*.

## C.3 Network Structure

The network structure of our model is illustrated in Fig. 6. In general, we adopt the same model architecture as in DiffBIR [27], using LR images as conditional inputs for the ControlNet module. The ControlNet is initialized from the Stable Diffusion 2.1 Unet encoder and trained to generate HR images given LR inputs.

Table 6: Comparison of performance: w/o DoSG vs. different accelerated samplers on the *RealSR* and *DRealSR* datasets with same model(RealESRNet preprocessing + DiffBIR). In all setups, inference is carried out over 5 steps.

| Method | Corr. Sampler | RealSR Dataset | | | | DrealSR Dataset | | | |
|--------|---------------|---------|--------|---------|--------|---------|--------|---------|--------|
| | | CLIPIQA↑ | MUSIQ↑ | MANIQA↑ | TOPIQ↑ | CLIPIQA↑ | MUSIQ↑ | MANIQA↑ | TOPIQ↑ |
| w/o DoSG (DDPM) | DDIM($\eta = 1$) | 0.5176 | 57.95 | 0.4293 | 0.5286 | 0.4732 | 48.22 | 0.3518 | 0.4316 |
| | EDM | 0.5351 | 62.08 | 0.4445 | 0.5789 | 0.5341 | 53.13 | 0.3917 | 0.5082 |
| | DPM-Solver++ -3 | 0.5323 | 62.67 | 0.4384 | 0.5807 | 0.5379 | 54.09 | 0.3932 | 0.5180 |
| w/ DoSG (DoSSR) | DoS SDE-Solver -1 | 0.6874 | 66.55 | 0.5574 | 0.6588 | 0.5907 | 59.12 | 0.4686 | 0.5907 |
| | DoS SDE-Solver -2 | **0.7025** | 69.27 | **0.5794** | 0.6966 | 0.6749 | 64.09 | 0.5196 | 0.6571 |
| | DoS SDE-Solver -3 | **0.7025** | **69.42** | **0.5781** | **0.6985** | **0.6776** | **64.40** | **0.5214** | **0.6618** |

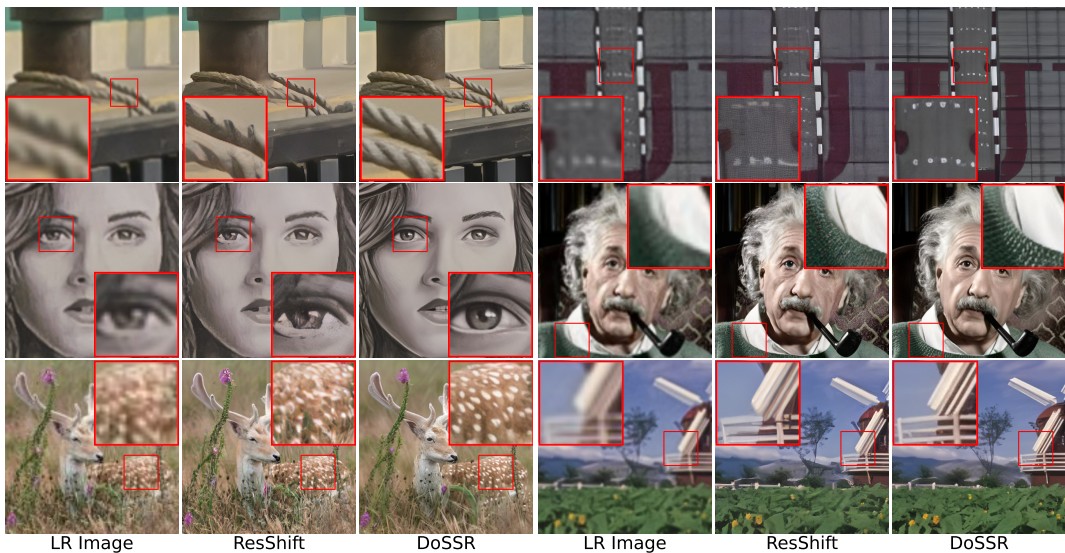

Figure 5: Qualitative comparisons between our retrained DoSSR and ResShift. The utilization of the diffusion prior noticeably enhances the realism and visual appeal of the generated high-resolution images. Please zoom in for a better view.

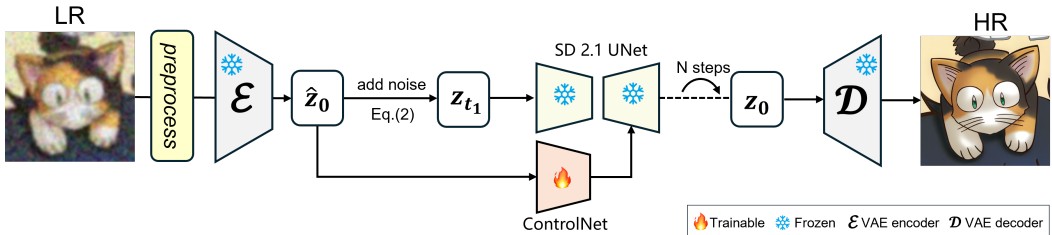

Figure 6: The overall framework of DoSSR. During training, we introduce noise to facilitate the gradual transition from the HR to LR domain, integrating it with the standard diffusion process, and incorporate preprocessed LR as a conditioning input for the denoising process, following the ControlNet approach. During inference, we add noise to LR latent according to Eq. (2) and perform inference starting from $t_1$.

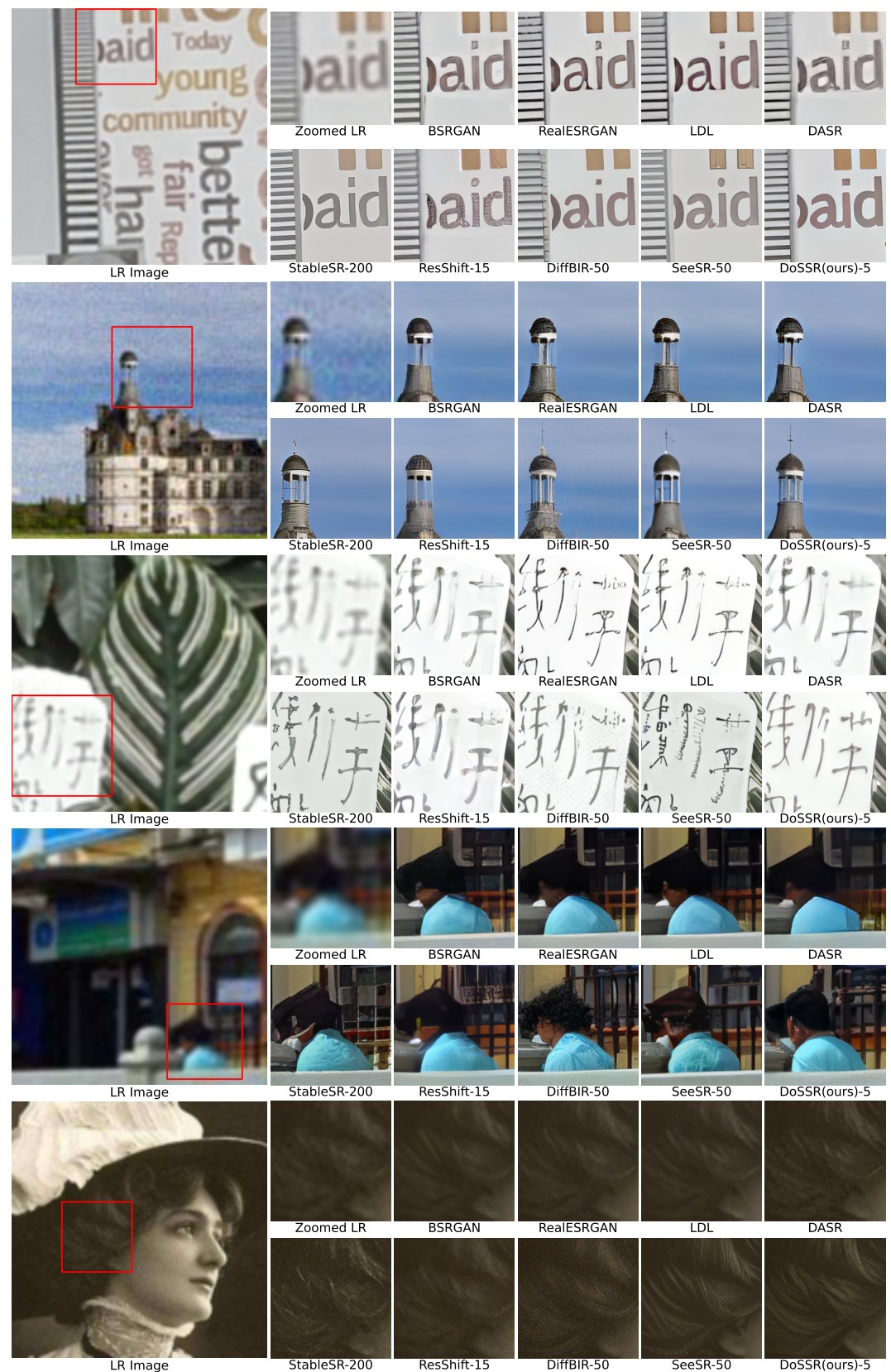

Figure 7: Qualitative comparisons of different steps of our DoSSR and other diffusion-based SR methods. The suffix "-N" appended to the method name indicates the number of inference steps. Please zoom in for a better view.

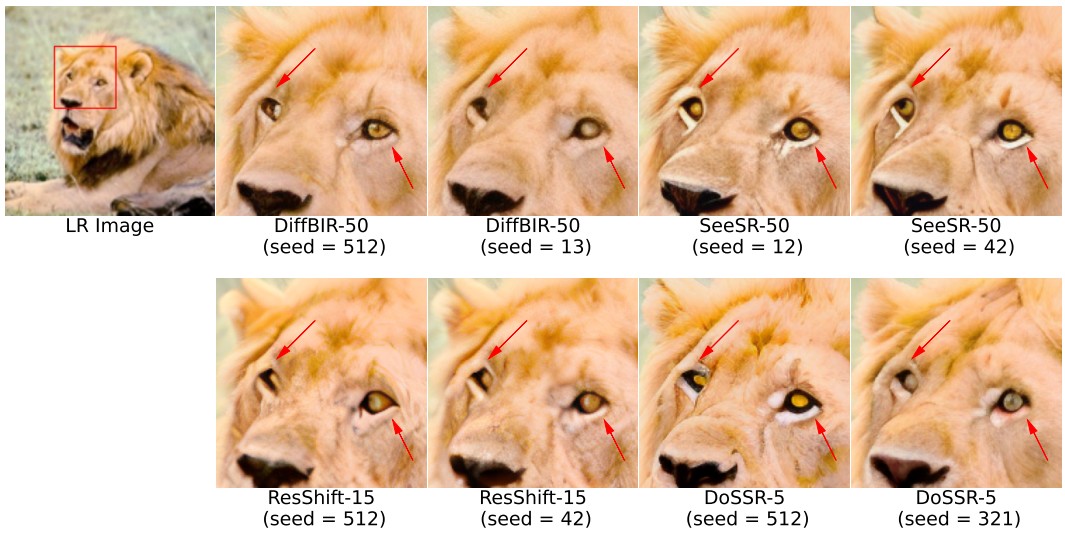

Figure 8: Visualizing the impact of random seed on diffusion-based methods. The "-N" suffix denotes inference steps. Please zoom in for a better view.

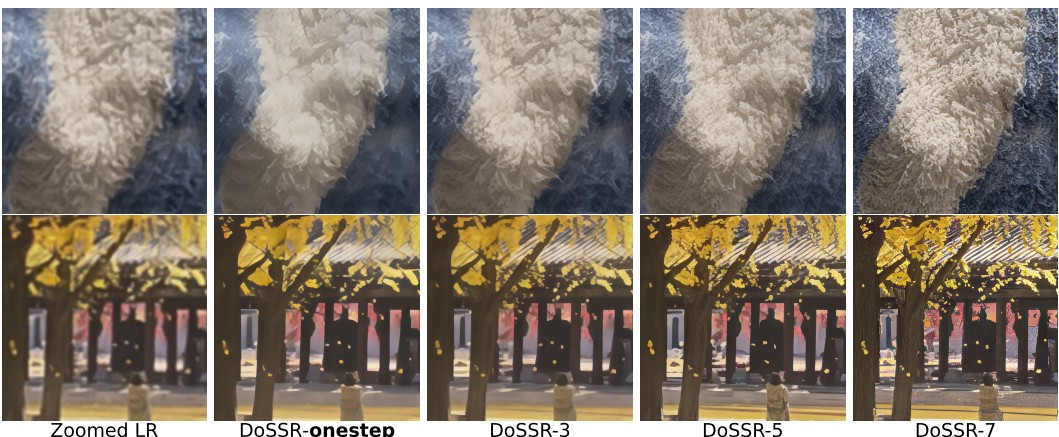

Figure 9: Qualitative comparisons of different inference steps of our DoSSR. The "-N" suffix denotes inference steps. Please zoom in for a better view.

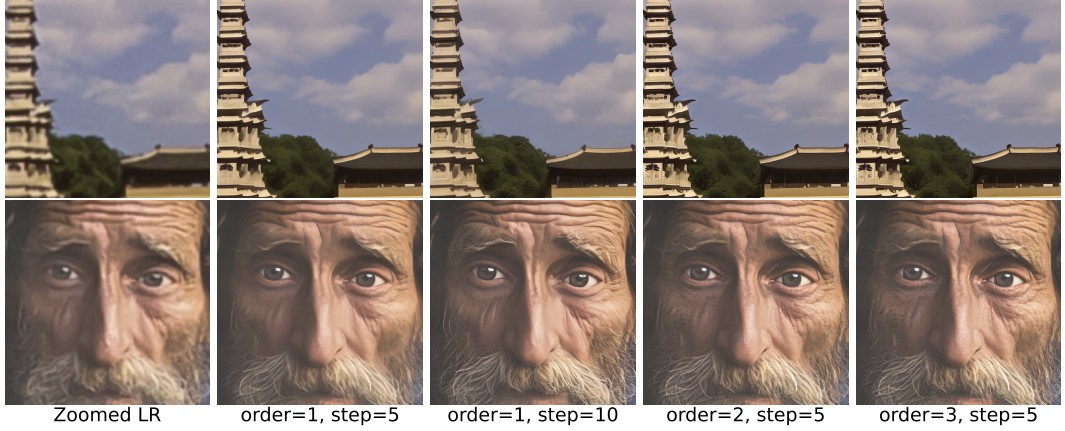

Figure 10: Qualitative comparisons of different sampler orders of our DoSSR. Please zoom in for a better view.

