# OpenReview forum: "Taming Diffusion Prior for Image Super-Resolution with Domain Shift SDEs"
_NeurIPS.cc/2024/Conference — NeurIPS 2024 poster_

### Official Review · Reviewer_dM7n · 2024-07-09

**Soundness:** 3
**Presentation:** 3
**Contribution:** 3
**Rating:** 6
**Confidence:** 5

**Summary:**

This paper proposed a Domain Shift diffusion-based SR model, namely DoSSR, which capitalizes on the generative powers of pretrained diffusion models while significantly enhancing efficiency by initiating the diffusion process with low-resolution (LR) images. In addition, authors advance their method by transitioning the discrete shift process to a continuous formulation to further enhance sampling efficiency. The proposed method is evaluated on synthetic and real-world datasets and achieved competitive results.

**Strengths:**

1. The proposed DoS-SDEs achieve a remarkable speedup in inference, demonstrating its superior efficiency.
2. The proposed method achieved competitive results on non-reference metrics.
3. The paper is well-organized and the figure is clear to understand.

**Weaknesses:**

1. The Domain Shift scheme seems to be similar to ResShift, where noise schedule is replaced with the solution in DDPM. Can the authors further explain the novelty of proposed Domain Shift scheme?
2. The experimental setting of the paper is insufficient. See questions for details.

**Questions:**

1. The method proposed in the paper is based on the RealESRGAN preprocessing and LAControlNet (from DiffBIR). Is the model trained based on the pre-trained model of DiffBIR? What about the results on the baseline1 "RealESRGAN preprocessing + DiffBIR" or baseline2 "RealESRGAN preprocessing + w/o Domian Shift Guidance(DoSG)"? Are there some experiments to verify the performance improvement of the proposed method compared to the baseline1 or baseline2?
2. What are the advantages of the DoSSR Solvers proposed in the paper compared to existing accelerated sampling schemes (such as DDIM [1], Dpm-solver++ [2], EDM [3])? Are there any ablation studies to verify it?

The relevant references are listed as follows,

[1] Song, Jiaming, Chenlin Meng, and Stefano Ermon. "Denoising diffusion implicit models." arXiv preprint arXiv:2010.02502 (2020). \
[2] Lu, Cheng, et al. "Dpm-solver++: Fast solver for guided sampling of diffusion probabilistic models." arXiv preprint arXiv:2211.01095 (2022). \
[3] Karras, Tero, et al. "Elucidating the design space of diffusion-based generative models." Advances in neural information processing systems 35 (2022).

**Limitations:**

The authors fully explain the limitations and potential societal impact in this paper.

---

> ### Author Rebuttal · Authors · 2024-08-07
>
> **Q1: Novelty against ResShift**
>
> **A1:**  Conceptually, our method is similar to ResShift, as both involve the design of a diffusion bridge between the LR and HR images.
> The main differences between the two diffusion schemes are that ours are carefully designed in order to better leverage the generative prior of pretrained diffusion models.
> Specifically, this results in different design chioces of the interpolation schedule and the noise schedule.
> A major advantage that our formulation brings is that we achieve a better trade of between generative quality and inference efficiency.
> Furthermore, we extend our formulation from discrete version to a continuous version and propose a  solver based on  stochastic differential equation (SDE). This solver improves our inference efficiency even furhter.
>
> We refer the reviewer to see more comparisons between ResShift and our method in our general response, Appendix A.7 and Appendix C.1.
>
> ***
>
>  **Q2: More baseline results**
>
> **A2:**  Fisrt, our model is NOT trained based on the pre-trained model of DiffBIR. Instead, we use the DiffBIR architecture and initialize the weights with Stable Diffusion 2.1 weights. Specifically, our baseline should be the baseline2 as you mentioned: "RealESRGAN preprocessing + w/o Domian Shift Guidance(DoSG)".
>
> The results of the baseline2 here can be found in Table 1 in the attached PDF, where this baseline is represented as "w/o DoSG (DDPM)". We compare the baseline with ours under 3 different samplers with varied orders. All results show our method outperforms the baseline by a large margin.
>
> ***
>
>  **Q3:Relation to existing solvers**
>
> **A3:**  Our proposed sampler, DoS SDE-Solver, is specifically designed for the model under our forward-noise scheme framework. Existing samplers, such as DDIM, DPM-Solver, and EDM, are not suitable for our model as they are designed for the default DDPM.  Therefore these sovlvers cannot be adapted to our model without bells and whistles.
>
> However, we can still show some analogy comparisons by comparing DDIM to the 1st order version of the DoS SDE-Solver, EDM to the 2nd order version, and 3rd order DPM-Solver++ to our 3rd order version.
>
> In fact, there are some interesting mathematical connections between our proposed DoS SDE-Solver and DPM-Solver series. When $\eta_t \equiv 0$ ,our DoS SDE-Solver Eq.(12) becomes
> $$
> x_t = \frac{\alpha_t}{\alpha_s}\frac{\lambda_t^2}{\lambda_s^2}x_s+\alpha_t(1-\frac{\lambda_t^2}{\lambda_s^2})x_\theta(x_s,\hat{x}_0,s)+\alpha_t\sqrt{\lambda_t^2-\frac{\lambda_t^4}{\lambda_s^2}}z_s, \lambda_t = \frac{\sigma_t}{\alpha_t(1-\eta_t)}
> $$
> the DoSG introduced by us becomes ineffective, and our DoS SDE-Solver is equivalent to the SDE DPM-Solver++[1],
> $$
> x_t=\frac{\sigma_t}{\sigma_s}e^{-h}x_s + \alpha_t (1-e^{-2h}) x_θ(x_s,s) +\sigma_t\sqrt{1-e^{-2h}}, h = log(\alpha_t/\sigma_t)-log(\alpha_s/\sigma_s).
> $$
>
> **We show quantitative results in Table 1 in the attached PDF**. In all setups, inference is
> carried out over 5 NFEs(number of function evaluations) with roughly same latency . From Table 1, two conclusions can be drawn: Firstly, regardless of DDPM or DoSSR, higher orders lead to better performance. Secondly, our DoSSR demonstrates superior performance compared to the corresponding order solver with DDPM, benefiting from the inclusion of DoSG in our DoS SDE-Solver.
>
> ***
>
> Reference:
>
> [1] Lu C, Zhou Y, Bao F, et al. Dpm-solver++: Fast solver for guided sampling of diffusion probabilistic models[J]. arXiv preprint arXiv:2211.01095, 2022.

---

> > ### Comment · Reviewer_dM7n · 2024-08-10
> > **Follow-up Feedback**
> >
> > Thank you for your respond. However, I still have some concerns about the real application value of the proposed DoS SDE-Solver. As the author said, "Our proposed sampler, DoS SDE-Solver, is specifically designed for the model under our forward-noise scheme framework" and "Existing samplers, such as DDIM, DPM-Solver, and EDM ...  cannot be adapted to our model". Therefore, I doubt whether DoS SDE-Solver can be easily extended to other models or tasks, and whether it will bring a large training cost.

---

> > > ### Author Response · Authors · 2024-08-11
> > >
> > > Thanks for the quick feedback! We here further address the concern regarding the practical application value of the proposed DoS SDE-Sovler. This can be explained from the following two perspectives.
> > >
> > > **1. Adapting Existing Solvers to the DoS Diffusion Model:**
> > >
> > > While it is feasible to apply solvers like DDIM, EDM, and DPM Solver to our DoS diffusion model, these adaptations require some effort in re-derivation due to differences in the forward process. No additional training is necessary for this adaptation. For instance, the DDIM solver applied to the DoS diffusion model is equivalent to the first-order version of our DoS SDE-Solver, while the DPM Solver++ corresponds to the third-order version. Although we haven't shown an exact equivalence for the EDM solver due to differing noise and scaling schedules, it is analogous to the second-order version of our solver. In conclusion, adapting DDIM, EDM, and DPM-Solver for the DoS diffusion model is unnecessary, as our DoS SDE-Solver already provides a comprehensive solution.
> > >
> > > As a summary, we do not need adapting DDIM, EDM and DPM-Solver for the DoS diffusion model, because the proposed DoS SDE-Solver already provides a unified solution.
> > >
> > > **2. Applying DoS SDE-Solver to other models and tasks:** It is also feasible to apply our solver to other models and tasks.
> > >
> > > -  Different models:  if we apply it to a DDPM pretrained model (such as the Stable Diffusion series), we do need additional finetuning. However, we do not see this as a defect. Diffusion-based image SR methods ALWAYS necessitate finetuning for good performance. For example, in our baseline model, DiffBIR[1], the finetuning cost is over 384 A100 GPU hours(2~3 days on 8x A100), and in StableSR[2], it costs about 960 v100 GPU hours(5 days to one week on 8x V100 GPUs). In contrast, our finetuning cost is relatively low, at just ~192 v100 GPU hours (1 day on 8x V100), delivering state-of-the-art super-resolution quality and a 5x-7x speedup during inference.
> > >
> > > - Different tasks:  The DoS SDE-Solver can also be extended to image-to-image translation tasks, including image restoration (e.g., deblurring, deraining) and style transfer. These tasks fit within the domain shift framework, requiring only paired image data to replace the LR-HR pair used in our super-resolution setting.
> > >
> > > We hope this response addresses your concerns. Please feel free to reach out with any further questions.
> > >
> > > ***
> > > References:
> > >
> > > [1] Lin X, He J, Chen Z, et al. Diffbir: Towards blind image restoration with generative diffusion prior[J]. arXiv preprint arXiv:2308.15070, 2023.
> > >
> > > [2] Wang, Jianyi, et al. "Exploiting diffusion prior for real-world image super-resolution." International Journal of Computer Vision (2024): 1-21.

---

### Official Review · Reviewer_tfJj · 2024-07-12

**Soundness:** 2
**Presentation:** 3
**Contribution:** 2
**Rating:** 5
**Confidence:** 5

**Summary:**

The existing diffusion-based image restoration models do not fully exploit the generative prior of the pre-trained diffusion models and produce high-quality images starting from Gaussian noise, leading to inefficiency. To address these issues, the paper a Domain Shift diffusion-based SR model (DoSSR) which can leverage the generative power of the pre-trained diffusion models and reconstruct super-resolved images based on the input corrupted images. It shows that the proposed method can effectively improve efficiency with few evaluation steps. Specifically, experiment results show that the proposed model achieves a slightly better performance on some benchmark datasets, including synthesized data and real-world data, in terms of various perceptual quality measurements.

**Strengths:**

1. The paper clearly describes the limitations of the existing diffusion-based image restoration models, which are hot topics in image restoration based on Gaussian diffusion models. The structure of the paper is well-organized, so the whole paper is understandable and easily follow.

2. How to efficiently perform image restoration based on the diffusion models is a significant challenge. The paper demonstrates that it can achieve better performance using few evaluation steps, which is impressive.

3. The paper evaluates the proposed method on several benchmark datasets, which contain synthesized data and real-world data, and the ablation studies are comprehensive.

**Weaknesses:**

1. In implementation, this research adopts LAControlNet and SD 2.1-base as the pre-trained T2I model. However, the paper does not illustrate the overall structure of their proposed method, so it is hard to understand how to adopt the corrupted input as conditional information.

2. The novelty of the paper is limited. 1). Leveraging pre-trained large-scale diffusion models for image super-resolution under the framework of ControlNet has been investigated in many previous studies [1-4]. These methods can also produce high-quality restored images from the corrupted input. 2). The goal of the proposed domain shift SDE  is to design a diffusion bridge between the corrupted input images and the corresponding high-quality images, which is very similar to flow matching-based methods, such as rectified flow [5]. However, the paper does not discuss the flow-based models.

3. By leveraging flow-based models, several methods [6-7] have been proposed for image restoration.

4. The performance improvement on several datasets is limited. It is hard to evaluate the visual quality of images generated by the proposed method through visual comparison because no GT image is illustrated.


[1] Wang J, Yue Z, Zhou S, Chan KC, Loy CC. Exploiting diffusion prior for real-world image super-resolution. IJCV, 2024.

[2] Chen X, Tan J, Wang T, Zhang K, Luo W, Cao X. Towards real-world blind face restoration with generative diffusion prior. IEEE Transactions on Circuits and Systems for Video Technology. 2024 Apr 1.

[3] Yu F, Gu J, Li Z, Hu J, Kong X, Wang X, He J, Qiao Y, Dong C. Scaling up to excellence: Practicing model scaling for photo-realistic image restoration in the wild. In CVPR 2024.

[4] Yang T, Ren P, Xie X, Zhang L. Pixel-aware stable diffusion for realistic image super-resolution and personalized stylization. arXiv preprint arXiv:2308.14469. 2023.

[5] Liu X, Gong C, Liu Q. Flow straight and fast: Learning to generate and transfer data with rectified flow. ICLR, 2023.

[6] Liu J, Wang Q, Fan H, Wang Y, Tang Y, Qu L. Residual denoising diffusion models. CVPR, 2024.

[7] Zhu Y, Zhao W, Li A, Tang Y, Zhou J, Lu J. FlowIE: Efficient Image Enhancement via Rectified Flow. CVPR, 2024.

**Questions:**

1. It is better to illustrate the overall structure of the adopted models. For example, how to adopt the corrupted input as conditional information for the T2I model.
2. The novelty of the paper is a significant concern. It is better to demonstrate the advances of the proposed method, compared with the previous studies.
3. The paper proposes to specifically design the drift term in SDE for image super-resolution, which leads to the reduction of evaluation steps. It easily causes confusion, because the additional drift term is to interpolate the corrupted input and the corresponding high-quality images. The sampling method is the key point. These two points should be clarified.

**Limitations:**

The paper does not discuss the limitations and social impact of the proposed method. It is hard to evaluate the model fairness, generalization ability, robustness, and transparency of the proposed method.

---

> ### Author Rebuttal · Authors · 2024-08-07
>
> **Q1: Overall model structure and implementation details**
>
> **A1:**  Thanks for the valuable suggestion! Please find the figure and descriptions about detailed model structure in the attached PDF.  Also, Figure 2(a)(b) in the main text shows training and inference pipelines and may help better understand.  Generally, we adopted the same model structure as in DiffBIR[1], where we use LR as condition input for the ControlNet module. The ControlNet here is initialized from the Stable Diffusion 2.1 Unet Decoder, then trained to generate HR images given LR images as condition inputs. We will add these to our appendix in the next version.
>
> ***
>
> **Q2: Novelty against ControlNet based methods**
>
> **A2:**  Yes, our method follows the ControlNet framework similar to the listed literature. However, we want to emphasize that we did not claim the ControlNet framework as our contribution. Rather, the only reason that we opt to it  is that we adopt DiffBIR, which is based on ControlNet,  as our baseline. It is totally possible to implement our method without ControlNet, since our innovation is mainly on the formulation of domain shift diffusion and a corresponding efficient sampler.
>
> The effecitveness of our proposed DoSSR can be justified by comparing the DiffBIR baseline with ours, please see Table 1 in the main text. The comparison is fair because the ControlNet part is exactly the same, and the only difference is that we  replace the learning objective and sampler from DDPM/DDIM with our propsed ones. We see a significant boost in generation quality, with $+6.9$\% MUSIQ and $+27.4$\% MANIQA in RealSR test set. Meanwhile, The inference efficiency is largely improved also, with number of function evaluations (NFE) decreased from 50 to 5, and lantency reduced by 5x from 5.85s to 1.03s.
>
> We would like to once again emphasize that our contribution is not the ControlNet-based SR framework. Please see our general response for a better illustration of our contributions.
>
> ***
>
> **Q3: Novelty against Rectified Flow**
>
> **A3:**  Thanks for the suggestion. Rectified Flow[2] (or we take the implementation of FlowIE[3] as an example for the SR task) does share a similar spirit with our formulation, from the pespective of the LR-to-HR interpolation design. However, our work differs from Rectified Flow mainly on that we focus more on better leverage of DDPM pretrained generative prior. Specifically, this results in different design chioces of the interpolation schedule and the noise schedule. Finally, we show under a fair comparison setting, our method remarkbly outpuerforms FlowIE, thanks to the better leverage of pretrained prior. Please see quantitative results in our general response. By the way, FlowIE is **concurrent with our work**, as it was not publicly available prior to the NeurIPS submission deadline.
>
> ***
>
> **Q4：Regarding the visual comparison**
>
> **A4:** For real-world image super-resolution tasks, since the ground truth represents just one of many plausible results, many prior arts do not include the GT[1,4,5], and we adhere to this practice as well. However, we appreciate the suggestion and will include a comparison with the ground truth in future versions to better present the visual comparison.
>
> ***
>
> **Q5：Clarification on the drift term of DoS-SDE solver**
>
> **A5:** Thanks for the suggestion. We believe this is a misunderstanding. In fact, the interpolation between LR and HR only exists in the training process. During inference, the sampler does not require an interpolation. Please note in Eq (12) the drift term (Domain Shift Guidance) contains only the LR latent $\hat{x}_0$.
>
> ***
> Finally, regarding the discussion of the limitations and social impact of the proposed method, due to page constraints, we have placed it in Appendix C.2. We intend to move it to the main body in the next version.
>
> ***
> References:
>
> [1] Lin X, He J, Chen Z, et al. Diffbir: Towards blind image restoration with generative diffusion prior[J]. arXiv preprint arXiv:2308.15070, 2023.
>
> [2] Liu X, Gong C. Flow Straight and Fast: Learning to Generate and Transfer Data with Rectified Flow[C]//The Eleventh International Conference on Learning Representations.
>
> [3] Zhu Y, Zhao W, Li A, et al. FlowIE: Efficient Image Enhancement via Rectified Flow[C]//Proceedings of the IEEE/CVF Conference on Computer Vision and Pattern Recognition. 2024: 13-22.
>
> [4] Sun, Haoze, et al. "Coser: Bridging image and language for cognitive super-resolution." _Proceedings of the IEEE/CVF Conference on Computer Vision and Pattern Recognition_. 2024.
>
> [5] Yu, Fanghua, et al. "Scaling up to excellence: Practicing model scaling for photo-realistic image restoration in the wild." _Proceedings of the IEEE/CVF Conference on Computer Vision and Pattern Recognition_. 2024.

---

> > ### Comment · Reviewer_tfJj · 2024-08-11
> >
> > thank you for your responses and comprehensive explanation. Your responses have addressed my concern and the final score has been raised to borderline accept.

---

### Official Review · Reviewer_7zor · 2024-07-12

**Soundness:** 3
**Presentation:** 3
**Contribution:** 3
**Rating:** 6
**Confidence:** 4

**Summary:**

This paper proposes a Domain Shift diffusion-based SR model, named DoSSR, that capitalizes on the generative powers of pretrained diffusion models while significantly enhancing efficiency by initiating the diffusion process with low-resolution (LR) images. Introducing the SDEs to describe the process of domain shift and provide an exact solution for the corresponding reverse-time SDEs. Designing customized fast samplers, resulting in even higher efficiency, thereby achieving the state-of-the-art efficiency performance trade-off.

**Strengths:**

1.This paper is innovative and combines a domain shift equation with the existing diffusion model to obtain better inference results and efficiency.

2.This paper has a solid theoretical foundation, the author gives a strict and sufficient formula derivation process.

3.The fusion results obtained are very superior in the non-reference index.

**Weaknesses:**

1.	In Table 1, the results of the proposed method on the reference index seem unsatisfactory, even much lower than other diffusion model methods.

2.	In colddiff [r1], Bansal et al proposed a method of reasoning from an arbitrary starting point. What is the difference between the method of reasoning from LR mentioned in this article and it? Such clarification would be useful.

3.	In order to facilitate observation, it is recommended to perform a partial zoom on Fig. 4 (b) for closer examination.

[r1] Bansal, Arpit, et al. "Cold diffusion: Inverting arbitrary image transforms without noise." Advances in Neural Information Processing Systems 36 (2024).

**Questions:**

1. As mentioned in Weakness, why do the resulting graphs look very beautiful, but the reference indicators are so far behind?

2. Explain the differences with DPM-solver, since ito-taylor expansionin also exists in your paper.

---

> ### Author Rebuttal · Authors · 2024-08-07
>
> **Q1: Regarding the reference index.**
>
> **A1:**  For real-world image super-resolution tasks, using reference metrics is not always an effective way to measure the quality of the generated results, as the 'groundtruth' image is merely one of many plausible outcomes. For methods based on diffusion models, particularly those utilizing large-scale pretrained diffusion models, the models' strong generative capabilities often produce outputs that, while differing from the ground truth, remain reasonable. Consequently, this can lead to relatively low reference metrics, as similarly discussed in other papers [1,2,3].
>
> Additionally, the reference metrics of our method is reasonable as expected. Compared to our baseline model DiffBIR, we show improvements across three test datasets and is generally comparable to other SD-based models in Table 1.
>
> ***
>
> **Q2: Comparion to ColdDiff [4].**
>
> **A2:** The core idea of ColdDiff is to perturb the data distribution by sampling non-Gaussian noise (such as using blur, snow, or pixelation) to degrade it into a known data distribution.
> Taking the super-resolution task as an example, it allows for reasoning from an arbitrary starting point because LR images can be approximated as a node in the forward degradation process.
> Thus, one can initiate the reverse process from this node to perform degradation inference.
> However, doing so introduces a gap between training and inference, because the degradation is hand-crafted in training, differing from real-world degradations. This will lead to a significant drop in model performance.
>  In contrast, our method, through a carefully designed forward diffusion equation, allows for precise calculation of the insertion inference from the intermediate node, as the latter half of the forward process is seen in the training process.
>
> Another drawback of ColdDiff is that it is designed on the pixel space, while modern diffusion models mostly operate on a latent space. Degradation in the pixel space is not always equivalent to degradation in the latent space, makes it difficult for adaptation to latent diffusion models .
>
> We also provide experimental results in the **general response**, comparing our method to ColdDiff. It can be seen even using the full sample scheme (not skipping the first half steps), ColdDiff largely underperforms ours.
>
> ***
>
> **Q3: Regarding differences with DPM-solver.**
>
> **A3:** DPM-solver is derived under the DDPM framework. Since our forward diffusion process is different from that of DDPM, DPM-solver cannot be simply applied to our method. The proposed DoS-SDE solver is to our DoSSR as DPM-solver is to DDPM.
> The derivation of the two is similar as they both use ito-taylor expansion as a tool.
>
> In fact, there are some interesting mathematical connections between our proposed DoS SDE-Solver and DPM-Solver series. When $\eta_t \equiv 0$ ,our DoS SDE-Solver Eq.(12) becomes
> $$
> x_t = \frac{\alpha_t}{\alpha_s}\frac{\lambda_t^2}{\lambda_s^2}x_s+\alpha_t(1-\frac{\lambda_t^2}{\lambda_s^2})x_\theta(x_s,\hat{x}_0,s)+\alpha_t\sqrt{\lambda_t^2-\frac{\lambda_t^4}{\lambda_s^2}}z_s, \lambda_t = \frac{\sigma_t}{\alpha_t(1-\eta_t)}
> $$
> the DoSG introduced by us becomes ineffective, and our DoS SDE-Solver is equivalent to the SDE DPM-Solver++[5],
> $$
> x_t=\frac{\sigma_t}{\sigma_s}e^{-h}x_s + \alpha_t (1-e^{-2h}) x\_\theta(x_s,s) +\sigma_t\sqrt{1-e^{-2h}}, h = log(\alpha_t/\sigma_t)-log(\alpha_s/\sigma_s).
> $$
>
> ***
>
> **Q4: Zoom-in for better visualization.**
>
> **A4:** Thanks for the suggestion! We will include a partial zoom-in on Fig. 4(b) in the next version.
>
> ***
>
> References:
>
> [1] Wu, Rongyuan, et al. "Seesr: Towards semantics-aware real-world image super-resolution." _Proceedings of the IEEE/CVF conference on computer vision and pattern recognition_. 2024.
>
> [2] Wang, Jianyi, et al. "Exploiting diffusion prior for real-world image super-resolution." _International Journal of Computer Vision_ (2024): 1-21.
>
> [3] Yang, Tao, et al. "Pixel-aware stable diffusion for realistic image super-resolution and personalized stylization." _arXiv preprint arXiv:2308.14469_ (2023).
>
> [4] Bansal, Arpit, et al. "Cold diffusion: Inverting arbitrary image transforms without noise." _Advances in Neural Information Processing Systems_ 36 (2024).
>
> [5] Lu C, Zhou Y, Bao F, et al. Dpm-solver++: Fast solver for guided sampling of diffusion probabilistic models[J]. arXiv preprint arXiv:2211.01095, 2022.

---

> > ### Comment · Reviewer_7zor · 2024-08-10
> >
> > The authors' rebuttal solves my concerns, I raise my rating to WA.

---

### Official Review · Reviewer_QMdK · 2024-07-20

**Soundness:** 3
**Presentation:** 3
**Contribution:** 3
**Rating:** 5
**Confidence:** 5

**Summary:**

This paper presents DoSSR, a novel framework for diffusion-based image super-resolution that aims to balance efficiency and performance. A domain shift equation that integrates with existing diffusion models, allowing inference to start from low-resolution images while leveraging pretrained diffusion priors. A continuous formulation of the domain shift process using stochastic differential equations (SDEs), enabling fast and customized solvers.

**Strengths:**

1. The domain shift equation and SDE formulation provide a solution and enables efficient inference.
2. DoSSR outperforms existing methods on multiple benchmarks, both in terms of quality metrics and computational efficiency.

**Weaknesses:**

I have doubts about the interpolation between LR and HR. Theoretically, from the moment of interpolating, the image contains both the degradation of LR and the details of HR. For details, please refer to the article [Interpreting Super-Resolution Networks with Local Attribution Maps], which also mentions similar gradient and difference operations. If each image contains information about LR and HR, is it reasonable to use this for training? Based on the proposed method and the actual effect of LR&HR, I guess that using only LR or adding various noise/blur methods to LR may also achieve similar results. Since the topic of the paper is related to this approach, this concept is still important. In addition, skipping the first half of the diffusion model is a more intuitive method. I don’t know if there is a paper discussing this, but in practice this method seems to have appeared for some time (this will not affect my score much).

**Questions:**

About the issue mentioned in Weaknesses, can we have some experiments to show the reason of doing such interpolation?

**Limitations:**

yes

---

> ### Author Rebuttal · Authors · 2024-08-06
>
> **Q1: Regarding interpolation between LR and HR.**
>
> **A1:**  We want to clarify the necessity of interpolation between LR and HR from the following aspects.
> 1. _**HR is necessary:** All diffusion-based SR method consturct training samples using HR images._ This is because that our objective is to recover the HR images, thus the training samples are constructed by add noises on HR images.  The forward diffusion process can be seen as linear combinations (more general than interpolation) of HR and Gaussian noises.
>
> 2. _**LR is beneficial:** Using LR as starting point instead of Gaussian noise can shorten the diffusion path._ Imagine the original DDPM is to find a path from noise towards the HR domain, what we do by replacing noises with LR is to shorten the diffusion path by pushing forward the starting point from pure noise to the LR domain.
>
> 3. _**Interpolation is explored in existing works:**_  In fact we are not the first to introduce the interpolation diffusion process. As pointed by Reviewer tfJj and dM7n, the interpolation form has been explored in Rectified Flow [1], FlowIE [2], and ResShift [3].  Results of each work show good performance, and more importantly, better sampling efficiency. Though we share some similar spirits with these work in the interpolation formulation, our main contribution lies more on how to better leverage pretrained diffusion priors and introducing a novel fast solver. We suggest to read our **general response** for more comparisons.
> ***
>
> **Q2: Using only LR or adding various noise/blur methods to LR may also work?**
>
> **A2:**  Sorry we did not find an existing diffusion-based method that uses only LR to construct training samples, as by doing so it is only possible to recover the LR images theoritically.
> Similarly, adding various noise/blur methods to LR is not feasible, either. However, we do find a diffusion process, ColdDiff [4], that adds various blur to **HR** images and try to recover from the blured samples. In general, ColdDiff generates reasonable results, but significantly underperforms ours. Please see results in our general response and Reviewer 7zor Q2 for more details.
>
> Using only LR images as **conditions** rather than **inputs** can achieve super-resolution, which is the focus of comparative methods in our paper, such as StableSR [6] and DiffBIR [7]. However, these methods do not effectively address the trade-off between efficiency and performance, which is a key issue that our proposed method aims to resolve.
> ***
>
> **Q3: Comparison with methods can also skip half the inference steps.**
>
> **A3:** Indeed, there have been previous works implementing this idea, such as SDEdit [5]. However, their approach adopts an approximation that  $LR + noise \approx HR + noise$, thereby enabling inference to start from the intermediate node. Although this approximation can be used, it introduces a gap between training and inference, leading to a decrease in model performance. In contrast, our method meticulously designs the forward diffusion equation to enable precise calculation of this skip, thereby eliminating the training-inference gap and achieving excellent performance. We conducted a comparative experiment as shown in the table below:
>
> | Method   | from corrupted LR input | inference steps | MUSIQ$\uparrow$ | CLIPIQA$\uparrow$ | MANIQA$\uparrow$ |
> | ------ | :------------: | --------------- | --------------- | ----------------- | ---------------- |
> | StableSR | \                       | 200             | 65.88           | 0.6234            | 0.4260           |
> | StableSR | starting point  = 9T/10 | 180             | 55.45           | 0.4288            | 0.3314           |
> | StableSR | starting point  = T/2   | 100             | 23.40           | 0.2098            | 0.1873           |
> | DiffBIR  | \                       | 50              | 64.94           | 0.6448            | 0.4539           |
> | DiffBIR  | starting point  = 9T/10 | 45              | 50.09           | 0.4009            | 0.3176           |
> | DiffBIR  | starting point  = T/2   | 25              | 38.12           | 0.2751            | 0.2669           |
> | DoSSR    | starting point  = T/2   | 5               | **69.42**       | **0.7025**        | **0.5781**       |
>
> As seen from the table above,  the performance of other methods drops heavily when skipping inference steps.  Skipping half of the diffusion model (starting point from t = T/2) leads to a noticeable decline in the model's performance (-64.48\%  MUSIQ in StableSR , -41.29\%  MUSIQ in DiffBIR)  .  Skipping fewer steps (starting point from t = 9T/10)  still results in a performance decrease for the model ( -15.8\%  MUSIQ in StableSR, -22.87\%  MUSIQ in DiffBIR).
>
> ***
> Reference:
>
> [1] Liu, Xingchao, and Chengyue Gong. "Flow Straight and Fast: Learning to Generate and Transfer Data with Rectified Flow." The Eleventh International Conference on Learning Representations.
>
> [2] Zhu, Yixuan, et al. "FlowIE: Efficient Image Enhancement via Rectified Flow." Proceedings of the IEEE/CVF Conference on Computer Vision and Pattern Recognition. 2024.
>
> [3] Yue, Zongsheng, Jianyi Wang, and Chen Change Loy. "Resshift: Efficient diffusion model for image super-resolution by residual shifting." Advances in Neural Information Processing Systems 36 (2024).
>
> [4] Bansal, Arpit, et al. "Cold diffusion: Inverting arbitrary image transforms without noise." _Advances in Neural Information Processing Systems_ 36 (2024).
>
> [5] Meng, Chenlin, et al. "SDEdit: Guided Image Synthesis and Editing with Stochastic Differential Equations." _International Conference on Learning Representations_.
>
> [6]  Wang, Jianyi, et al. "Exploiting diffusion prior for real-world image super-resolution." International Journal of Computer Vision (2024): 1-21.
>
> [7]Xinqi Lin, Jingwen He, et al. Diffbir: Towards blind image restoration with generative diffusion prior. arXiv preprint
> 385 arXiv:2308.15070, 2023

---

> ### Comment · Reviewer_QMdK · 2024-08-13
> **Response to the rebuttal**
>
> I have read the author's response, as well as the comments and discussions with other reviewers. The author has partially addressed my concerns. I keep my init positive score.

---

### Author Rebuttal · Authors · 2024-08-06

# General Response
We thank all reviewers for their positive feedback:  solid theoretical foundation (Reviewer 7zor), outperforming existing methods on multiple benchmarks (Reviewer QMdK), better performance using few evaluation steps (Reviewer tfJj), and well-organized, clear to understand (Reviewer dM7n).
***

## **Regarding contributions / novelty.**

We here first want to clarify the main contributions of our work, especially in a way by comparing our method with other plausible alternative formulations of diffusion processes. The diffusion formulations that we compare with include:
- **ColdDiff [1] :** Similar to DDPM. The distinction is that it opts to other degradation forms, such as blur and masking, instead of additive Gaussian noises as in DDPM. In our case for image super-resolusion, we select the blur degradation. Note that ColdDiff is originally applied in the pixel space, while we have to implement it in the latent space for fair comparison with our method.

- **Rectified Flow [2]** defined the forward process by $x_t = t \hat{x}_0 + (1-t) x_0$ where $\hat{x}_0$ is the data sample and $x_0$ is Gaussian noise. In our case for image super-resolusion, we choose FlowIE[4] as an implementation of rectified flow which replaces $x_0$ with the LR image as the starting point.  For comparison, our forward pass is defined by $x_t = \alpha_t (\eta_t  \hat{x}_0 + (1-\eta_t) x_0 ) + \sigma_t$. The main differences are two-fold: First, the interpolation schedule is diffferent; Second, we introduce an outer degradation parameterized with $\alpha_t, \sigma_t$. Both differences are crucial for making it possible to  seamlessly adapt the DDPM pretrained weights to image super-resolution.

- **ResShift [3]**  defines the forward pass by $x_t = \eta_t \hat{x}_0 + (1- \eta_t) x_0 + k^2 \eta_t$. It can be seen as a weaken case of our formulation when we set $\alpha_t \equiv 1$ and $\sigma_t = k^2 \eta_t$, thus similar to our method.  However, the devil is in the details, our noise and scale factors are crucial for making it possible to effectively utilize the diffusion prior while ResShift fails.

We implement the above three forms of diffusion formulations under the exact same experiment setting as our DoSSR. Below we show the quantitative comaprison results on RealSR test set. More comprehensive results can be found in Table 2 in the attached PDF.
|  Method  |  MUSIQ$\uparrow$ | MANIQA$\uparrow$ | NFE$\downarrow$ |
| ------ |  :-----------------: | :-----------------: | :-----------------: |
| ColdDiff |  47.42           | 0.2783           | 5               |
| FlowIE |   50.82           | 0.3228           | 1               |
| ResShift |   56.01           | 0.4001           | 5               |
|  DoSSR   |   **62.69**       | **0.5115**           | 1             |
|  DoSSR   |  **69.42**       | **0.5781**           | 5             |

- **(1) Comparison to ColdDiff:** Results of ColdDiff is significantly worse than other methods. The main reason can be summerized as two aspects. First, applying bluring kernels to latent features is not equivalent to applying them to raw images, while ColdDiff is originally designed for the latter. Second, the blurring kernel can only be designed in a  hand-craft manner and may not represent the real-world degradation, so when tested with real LR images there would be a domain gap. The limilation of hand-crafted degradation is also well-known in many previous image restoration works. In contrast, other methods, including our DoSSR, does not assume a fixed degradation, so the performance is much better.

- **(2) Comparison to FlowIE:** Because FlowIE emphasizes single step evaluation, we follow its setting and compare DoSSR with 1-setp evaluation with it.  It can be see that FlowIE is also not satisfactory and largely underperforms our DoSSR (-11.87\%  MUSIQ). This is mainly attributed to FlowIE's bad adaptation from the DDPM pretrained T2I model, since the learning objective of flow matching differs from score matching. By contrast, our design makes full use of the DDPM pretrained weights so performs much better.

- **(3) Comparison to ResShift:** ResShift also underperforms ours by a large margin. The reason is similar to why FlowIE underperforms ours: The formulation of ResShift does not consider adaptation from the pretrained diffusion prior. We suggest reviewers to see A.7 and C.1 in our Appendix for more detailed discussions.

To summarize, our formulation differs from other alternatives especially in terms of **better leverage and adaptation from DDPM pretrained T2I models**. Also, based on this formulation, we derive a fast SDE solver. These two designs leads to a very efficient solution for diffusion-based image SR, thus can be seen as our two main contributions over existing works.

Refereces:

[1] Bansal, Arpit, et al. "Cold diffusion: Inverting arbitrary image transforms without noise." Advances in Neural Information Processing Systems 36 (2024).

[2] Zhu, Yixuan, et al. "FlowIE: Efficient Image Enhancement via Rectified Flow." Proceedings of the IEEE/CVF Conference on Computer Vision and Pattern Recognition. 2024.

[3] Yue, Zongsheng, Jianyi Wang, and Chen Change Loy. "Resshift: Efficient diffusion model for image super-resolution by residual shifting." Advances in Neural Information Processing Systems 36 (2024).

[4] Zhu, Yixuan, et al. "FlowIE: Efficient Image Enhancement via Rectified Flow." Proceedings of the IEEE/CVF Conference on Computer Vision and Pattern Recognition. 2024.

---

### Decision · Program_Chairs · 2024-09-25

**Decision:**

Accept (poster)

**Comment:**

The rebuttal has well addressed the raised issues. All the reviewers recommend to accept this paper. The area chairs agree with this recommendation.